



# Halo ratio from ground based all-sky imaging

Paolo Dandini[1], Zbigniew Ulanowski[1], David Campbell[1], and Richard Kaye[2]

[1]School of Physics Astronomy and Mathematics, University of Hertfordshire, Hatfield, AL10 9AB, UK
[2]School of Engineering and Technology, University of Hertfordshire, Hatfield, AL10 9AB, UK

*Correspondence to:* Z. Ulanowski (z.ulanowski@herts.ac.uk)

**Abstract.** The halo ratio (HR) is a quantitative measure characterizing the occurrence of the 22° halo peak associated with cirrus. We propose to obtain it from the scattering phase function (SPF) derived from all-sky imaging. Ground based fisheye cameras are used to retrieve the SPF by implementing the necessary image transformations and corrections. These consist of geometric correction of lens distortion by utilizing positions of known stars in a camera image, transforming the images from the zenith-centred to the light-source-centred system of coordinates, correcting for the air mass and for vignetting, the latter using independent measurements from a sun photometer. The SPF is then determined by averaging the image brightness over the azimuth angle and the HR by calculating the ratio of the SPF at two scattering angles in the vicinity of the 22° halo peak. In variance from previous suggestions we select these angles to be 20° and 23°, on the basis of our observations. HR time series have been obtained under various cloud conditions, including halo cirrus, non-halo cirrus and scattered cumuli. While the HR measured in this way is found to be sensitive to the halo status of cirrus, showing values typically >1 under halo producing clouds, similar HR values, mostly artefacts associated with bright cloud edges, can also be occasionally observed under scattered cumuli. Given that the HR is an ice cloud characteristic, a separate cirrus detection algorithm is necessary to screen out non-ice clouds before deriving reliable HR statistics. Here we propose utilizing sky brightness temperature from infrared radiometry: both its absolute value and the magnitude of fluctuations obtained through detrended fluctuation analysis. The brightness temperature data permits the detection of cirrus in most but not all instances.

## 1 Introduction

Cirrus clouds are composed of ice crystals. It is well established that because of their high global coverage their impact on the Earth's climate is significant and to quantify it the microphysical and radiative properties of cirrus have to be better represented in atmospheric models (Baran, 2012). This is not trivial as the ice crystals which compose cirrus can take on a wide variety of non-spherical shapes and have size ranging from a few microns up to over a millimetre, making detailed characterization of cirrus difficult and light scattering by cirrus highly challenging to model. Furthermore, as cloud forcing must be quantified from solar to thermal wavelengths, a correct parameterization of cirrus properties is necessary over the same spectrum. To characterize cirrus we propose here the use of sky imaging.

Sky imaging finds application in determining fractional cloud cover (Johnson and Hering, 1987; Long and DeLuisi, 1998; Slater et al., 2001; Long et al., 2001; Berger et al., 2005; Kassianov et al., 2005; Cazorla et al., 2008b), macrophysical cloud



properties such as cloud brokenness, distribution, number and uniformity (Shields et al., 1997; Kegelmeyer, 1994; Long et al., 2006), in assessing the impact of cloud cover on surface solar irradiance (Pfister et al., 2003), in estimating cloud base height either from low-cost digital consumer cameras (Seiz et al., 2002; Janeiro et al., 2010) or by means of paired whole sky cameras (Lyons, 1971; Rocks, 1987; Allmen and Kegelmeyer, 1996), in cloud detection and classification (Calbó and Sabburg,

2008; Heinle et al., 2010; Ghonima et al., 2012), in short term weather forecasting (Chow et al., 2011), in characterizing aerosol (Cazorla et al., 2008a) and in determining cloud-free lines of sight (Shaklin and Lund, 1972; Shaklin and Lund, 1973; Lund, 1973; Lund et al., 1980). Sky imaging is not new in the field of cirrus investigation as testified by previous works on measurements of the 22° halo intensity from photographic photometry (Lynch et al., 1985), on the effects of ice-crystal structure on halo formation (Sassen et al., 1994) and on the characterization of cirrus through a combination of polarization

lidar and photographic observations of cirrus optical displays (Sassen et al., 2003). Yet sky imaging in itself is a poor technique for detecting cirrus, in particular when optically thin. Misdetection of thin clouds and limitations in cloud type classification are the major disadvantages of ground based sky cameras (Calbó et al., 2008; Heinle et al., 2010). This is true in particular within a scattering angle of 20° as forward scattering from thin cirrus or boundary layer haze and blooming of the camera sensor can give rise to artefacts that make the sky around the sun appear as if it is cloudy even if it is not (Tapakis et al.,

2013). A solution to this issue consists in using sun tracking occulting masks to prevent direct sun light from interfering with the image (Martinez-Chico et al., 2011). To overcome the problem of detecting cloud presence near the sun's location statistical approaches, using the mean and standard deviation of cloud coverage, have also been proposed (Pfister et al., 2003; Long, 2010). One of the first thin cloud algorithms to be developed (Shield et al., 1990), was based on the ratio Red/Blue (R/B) threshold method nowadays commonly adopted in most of the algorithms processing data from sky cameras and used

to distinguish clear from cloudy sky pixels. Field images, a red, a blue and a red and a blue trimmed with neural density were calibrated for the non-linearity of the basic sensor and for differences in the pass bands of the spectral filters and then used to obtain the corresponding blue to red ratio images before a threshold was set for determining the presence of thin clouds. It was found that uniform thin clouds gave rise to a significant increase in this ratio, lending themselves to detection. In a more recent version of the same algorithm (Shields et al., 2013) thin cloud detection is based on a haze/aerosol-corrected clear sky

NIR/blue ratio image. The algorithm automatically corrects for aerosol/haze variations and hardware artefacts. Progress has been made over the years with the development of particularly promising algorithms for cirrus detection based on the use of polarizing filters (Horvath et al., 2002). Moreover it is speculated that cloud texture, the standard deviation of cloudy pixel brightness, lends itself to distinguishing light-textured cirrus from heavier-textured clouds such as cumuli (Long et al., 2006). However, the Red/Blue (R/B) threshold method and its improved version, the Red-Blue difference threshold technique, can fail

in detecting cirrus (Heinle et al., 2010).

Nevertheless, sky imaging is effective for recording the optical displays sometimes associated with cirrus called halos and for measuring the angular distribution of scattered light. From the scattering phase function (SPF) the halo ratio (HR) is then calculated, which has previously been proposed as the ratio of the intensity of light scattered at 22° to the one at 18.5° (Auriol et al., 2001; Gayet et al., 2011), but was later obtained as the ratio of the average SPF between 21.5° and 22.5° to the

average between 18.5° and 19.5° (Ulanowski et al., 2014), or as the ratio of the maximum of light scattered at an angle $\theta_{max}$





between $21°$ and $23.5°$ to the minimum of light scattered between $18°$ and $\theta_{max}$ (Forster et al., 2017). HR is a quantitative measure of the strength of the $22°$ halo ring, occurring when randomly oriented, hexagonal columns refract light through facets inclined at $60°$ to each other. In this respect modelling studies of SPFs (Macke et al., 1996; Baran and Labonnote, 2007; Um and McFarquhar, 2010; Liu et al., 2013) associate the presence of halos mostly with highly regular crystals, although some

aggregates of smooth, regular prisms are also capable of producing halos (Ulanowski, 2005). Moreover, it has been shown on the basis of exact electromagnetic scattering techniques that the halo visibility implies the presence of large ice crystals with size parameter of the order of 100 or more (Mishchenko and Macke, 1998). Therefore the HR is an indirect measure of the size and regularity of the shape of the ice crystals forming the cloud. However, HR is also connected to cirrus reflectivity at solar wavelengths, in that the reflectivity is inversely proportional to the HR. This relates to other two important properties: ice

crystal roughness and the asymmetry parameter $g$, the average cosine of the scattering angle (Macke et al., 1996). The former is expected to be negatively correlated with the HR as rough ice crystal SPFs show enhanced back and side scattering. Modelling studies have estimated that the global-averaged SW cloud radiative effect associated with this enhancement due to ice particle surface roughness is of the order of 1-2 W m$^{-2}$ (Yi et al., 2013). In general roughening, internal inclusions and complex shapes are all major factors contributing to removing halo features from the SPFs (Shcherbakov, 2013). The asymmetry parameter on

the other hand is expected to be positively correlated with the HR (Ulanowski et al., 2006; Gayet et al., 2011; Ulanowski et al., 2014). Many studies (Korolev et al., 2000; Garrett et al., 2001; Baran and Labonnote, 2007; Shcherbakov et al., 2006; Gayet et al., 2011; Baum et al., 2011; Cole et al., 2013; Ulanowski et al., 2014; Baran et al., 2015) suggest that cirrus clouds are mainly formed by rough or complex particles giving rise to typically featureless SPFs, confirming indications from previous observations and explaining the relative rarity of ground-observed halo occurrences (Sassen et al., 1994). Nevertheless, recent

findings (Forster et al., 2017) suggest that the fraction of halo-producing cirrus might be larger than previously thought. This could be explained by modelling indicating that only a 10% fraction of smooth ice crystals is sufficient for the $22°$ halo display to occur. Less clear is the actual fraction of halo displays associated with preferentially oriented ice crystals. While Forster et al. (2017) observed more than 70% of total halo displays to be associated with oriented ice crystals, previous findings (Sassen et al., 2003) reported higher frequency of occurrence of $22°$ halos compared to sundogs and upper tangent arcs. To answer

these questions further long term observations are needed, and we propose and implement for this purpose a technique based on retrieving the HR through all-sky imaging.

The details and specifications of the all-sky cameras used in this investigation are given in section 2.2, the camera calibration is covered in section 2.2.1, the testing and correcting of the lens projection is discussed in section 2.2.2, the method used for determining the ice cloud SPF from images is covered in paragraph 2.3. Corrections for vignetting and for air mass are

described in sections 2.5 and 2.6, respectively. This is followed by detailed analysis of two test cases, where we also examine the issue of simultaneous detection of cirrus and its discrimination from warmer clouds.



## 2 Methods

### 2.1 Instrumentation

In addition to the all-sky cameras, the Cimel sun photometer CE318 N has been necessary to quantify image vignetting. The "Cimel" is a benchmark device for most aerosol observing networks and more specifically for the international federation of AERONET (Holben et al., 1998). Other observations, including brightness temperature (BT), measured through a narrow band infrared pyrometer (KT15.85 II, Heitronics) with spectral sensitivity peaking at about 10.6 μm, and solar irradiance, from pyranometer (SMP11, Kipp & Zonen B.V.) measurements have been used to confirm the presence of cirrus. The instrumentation was installed at the observatory at Bayfordbury (51.7748 N 0.0948 W) operated by the University of Hertfordshire.

### 2.2 The all-sky cameras

Two cameras were implemented, one set up for daytime, one for night-time observations. Both used the third version of the AllSky device, called the AllSky-340 (Santa Barbara Instrument Group, SBIG). Its optical system consists of a Kodak KAI-0340 CCD sensor and a Fujinon FE185C046HA-1 lens. The KAI-0340 is a VGA resolution (640x480 active pixels) 1/3" format CCD with 7.4 microns square pixels. The lens is of fish-eye type with a focal length of 1.4 mm and a focal ratio range of f/1.4 to f/16, fixed open at f/1.4 in the night camera and closed down to f/16 in the daytime camera. This combination gives a field of view of $185° \times 144°47'$ and average resolution of $18'$ per pixel. Assuming the camera is aimed at the zenith, a maximum of 95.0% of the area of the sky can be imaged, with 2.52% cut off at the top and bottom. The areas around the edges are most affected by light pollution and not suitable for any measurements so this loss is not too detrimental. During this study the cameras were located on the roof of a two-storey building (about 8 m above ground) where the elevated vantage point from the roof gives a clear view of the skies, uninterrupted by most of the surrounding features such as trees. The cameras are connected to a computer via an RS-232 serial cable. Using a serial-USB adapter at the PC end the maximum download speed of 460.8 kbaud can be achieved. At this rate the average download time of an image is 16.5 seconds for the gray camera. The colour one allows a download time of 48 seconds (Campbell, 2010).

The All-Sky cameras are illustrated in Fig. 1. Both cameras are inside aluminium enclosures with acrylic domes protecting the fisheye lens. A specially developed occulting disk was mounted on the daytime camera to prevent stray light from affecting the imaging. It consists of an opaque disk, 10 cm in diameter, connected to an L-shaped rigid arm whose long and short sections are 60 cm and 20 cm, respectively. The long arm is mounted on a stepper motor that can adjust its elevation angle, in turn attached to a ring which fits around the camera enclosure and can be rotated using a second stepper motor. Time, date and location provided by a GPS module (GPS-622R, RF Solutions) is fed to a microcontroller (PIC18F46K22) that calculates the sun's position (azimuth and elevation), when the sun is above the horizon. The controller operates the motors via two ST L6472 stepper motor drivers. At sunset the disk is positioned slightly below the horizon and microcontroller then waits for the sun to rise again.





The colour camera has a Bayer mosaic used for filtering the red, blue and green wavelengths. The Matlab built-in function "demosaic" is applied to the raw image data to reconstruct the full colour image. The fish-eye lens enables sky observation over a field of view (FOV) as large as $187°59'$ along the ENE-WSW direction and as large as $142°38'$ along SSE-NNW. The FOV of the daytime camera along ENE-WSW is slightly larger, $189°53'$. The difference is due to the different alignment of the lens relative to the sensor. This is also reflected in the position of the true zenith $(x_0, y_0)$ with respect to the centre of the camera plane, which for both cameras does not coincide with the true zenith. An offset of 14 pixels in the $x$ direction and 12 pixels in the $y$ direction that corresponds to about $4°$ and $3.5°$ respectively has been measured for the night time camera and about $1°$ and $9.5°$ for the daytime camera. The fish-eye lens employed in both cameras uses the "f-theta", or equidistant projection system which means that the distance in pixels from the true zenith of the object projected onto the camera plane is simply a scaling factor $f$ (here 3.365 pixels/degree) multiplied by the zenith angle $z$ of the object expressed in degrees (see section 2.2.1). The output images are available either in JPG or FITS format but the latter is preferred because of larger dynamic range and the absence of "digital development" correction.

### 2.2.1 Camera calibration

Geometric calibration of the camera is done by detecting the position of specific stars and planets in a night-time image and implementing a minimization procedure. Over the course of a clear night the daytime camera, whose aperture had been increased from the usual f/16 to f/1.4, was left to take images. Bright stars trajectories were plotted as curves on an image, based on their position in the middle of the exposure, the predicted movement and the theoretical f-theta system of the lens. The projection parameters of the camera were then adjusted until the errors between the calculated and true position of the stars on the image were minimized. The procedure will now be described in detail. First, the time the current image was taken was converted to the Julian date. Knowing this time and the longitude and latitude of the camera, the altitude and azimuth of the stars could be calculated from their right ascension and declinations (Meeus, 1998). These coordinates were transformed into the 2D plane of the image using the f-theta system, found to provide a good initial fit, then were shifted and rotated to account for the fact that the camera does not actually point precisely at the zenith, nor is the image top perfectly aligned with the true north. Finally the empirical scaling factor was applied to the coordinates in the $x$ and $y$ direction to compensate for the distortion created by the lens. This was found by plotting stars using the preliminary projection and determining the difference between the plotted stars and their corresponding background star. The errors as a function of the $x$, and then $y$ directions were fitted to a bi-cubic function and these added to the coordinates. The final result was found to align well to the background stars in the central parts of the image, and slightly less well around the edges. The equations which map astronomic coordinates into camera ones are:

$$x = fz\sin(A - \Delta) + x_0 \tag{1}$$

$$y = fz\cos(A - \Delta) + y_0 \tag{2}$$





where $(z, A)$ are the zenith and azimuth angles, $\Delta$ is the rotation of the camera from north, measuring about 16.4° and 13.6° for the night and day time cameras, respectively, $(x_0, y_0)$ are the pixel coordinates of the actual zenith and $f$ is the scale factor. The latter was found to be 3.365 px/deg. If $d$ is the pixel distance from zenith then the above equations can be rewritten as:

$$d = fz = \sqrt{(x - x_0)^2 + (y - y_0)^2} \tag{3}$$

### 2.2.2 Testing of the lens projection

In order to test the reliability of Eq. (1) and (2) in reproducing the actual mapping of the lens the same equations were used to track star and planet trajectories. On a clear night an image was acquired. Given $A$, $z$ and the acquisition time, pixel coordinates $(x, y)$ of a celestial object can be determined through Eq. (1) and (2). The actual location is detected by locating the bright spot

that corresponds to the star/planet. This is done by selecting an appropriate brightness threshold and converting the image to binary. From the binary image a square region centred where the star/planet is predicted to be $(x, y)$, and sufficiently large to include the spot, is then selected. The centre of mass of the spot then becomes the actual position. Fig. 2 shows predicted (blue dots) and actual positions (red dots) of several celestial objects on the night of the 15[th] of February 2013. The mean difference between predicted and actual position was mostly <0.1° aside from portions of Vega's, Deneb's, Arcturus', Sirius' and Rigel's

trajectories for which larger discrepancies, up to 0.38°, were observed. These relatively larger deviations are associated with the decreased accuracy of the method due to light pollution especially significant as the horizon is approached. For these reasons the lens projection was judged to be reasonably well replicated by the Eq. (1) and (2). Consequently, the following image transformations were implemented using exclusively the f-theta system.

### 2.3 Geometric transformations

The SPF is obtained by averaging the image brightness over pixels which are equidistant from the light source in terms of scattering angle. To achieve this, the raw image is transformed to move the light source to the zenith, to give a light-source centred system of coordinates. This is accomplished by mapping the original coordinates onto a sphere of unit radius and then by rotating the spherical coordinates to centre the light source at the origin of the final system. The first step consists in changing the projection from "linear" with respect to $z$ to proportional to sin $z$. Given $d$, from Eq. (4) $z$ is determined for each pixel $(x, y)$

and the new coordinates $(x', y')$ are calculated. The change of projection can be seen as the wrapping of the raw image around a sphere of unit radius with the constraint that the horizon, in the new coordinate system, will now be at a unit distance from the zenith. The periphery of the image which is below the horizon (see Fig. 3) is projected in the lower hemisphere while points above the horizon appear in the upper hemisphere. The projection of the raw image of Fig. 3 in the upper hemisphere is shown in Fig. 4. The new coordinates $x'$ and $y'$ determine the $Z$ coordinate: pixels above and below the horizon are associated with

positive and negative $Z$, respectively. A rotation transformation **T** is then used to rotate the coordinates $(x', y', Z)$ of a generic





pixel, identified by the vector $\boldsymbol{P}$, around the unit vector $u$ that is perpendicular to the line linking the true zenith and the light source, by an angle $\theta$ equal to the light source zenith angle (see Fig. 4) according to:

$$\mathbf{T}(\boldsymbol{P}) = (1 - \cos\theta)(\boldsymbol{P}\cdot\hat{u})\hat{u} + \cos\theta\boldsymbol{P} + \sin\theta(\boldsymbol{P}\times\hat{u}) \tag{4}$$

With the rotated coordinates (x'', y'', Z′) available, the transformed image in the upper hemisphere can be obtained, and an
example is shown in Fig. 5. This can be achieved by interpolating the original image into the new coordinates. Pixels that belong to those portions of the image that after rotation would be below the new horizon are in the lower hemisphere, which is not shown here. This should be accounted for if the SPF is to be calculated for all aky pixels and scattering angles available. Nevertheless that portion of the SPF is not necessary for HR calculation purposes and will not be covered here. The interpolated image is ultimately used to calculate the brightness as a function of the scattering angle. Fig. 6 (red-dashed line) shows the end
result corresponding to the raw image from Fig. 3.

## 2.4 Background mask

To prevent the contamination of the sky image by background objects above the horizon, a mask, derived from a summer time image, when vegetation is thicker, was used. To prevent the mask edge from falling within the region of the SPF (between 18° and 22°) from which the HR is derived, the measure was limited to images such that the light source $z < 65°$.

## 2.5 Vignetting correction

The falloff of brightness for increasing $z$, associated with vignetting, takes place in nearly every digital photograph, in all optical lens systems, in particular wide-angle and ultra wide-angle lenses like fisheye lenses (Jacobs and Wilson, 2007). In general, vignetting increases with the aperture and decreases with the focal length. Here it is quantified by comparing daytime image data with sun photometer data under clear sky. While the camera is affected by vignetting, the sun photometer is not, hence
the ratio of the corresponding radiance measures, over a similar spectral range, can be used to quantify vignetting. The blue channel of a clear sky daytime RGB image was extracted, being the best match to a spectral channel of the sun photometer. The image brightness along the principal plane (containing the zenith and the sun) and the corresponding sun photometer output were compared; by neglecting pixels such that $z$ is greater than approximately 20° over the meridian containing the sun, the average of the sun photometer sky brightness at the visible wavelengths of 0.5015 μm and 0.4403 μm was LOWESS smoothed
and fitted. Analogously a polynomial was fitted to the LOWESS smoothed image data. The ratio between the two fitting polynomials (camera to sun photometer) after normalization (see Fig. 7) is what we refer to as the devingetting coefficient (DC, see Fig. 8 black curve). As the DC was expected to be monotonically decreasing with $z$, the presence of a peak located roughly 8° from the zenith was investigated. Since the use of the occulting disk allows us to exclude stray light is the cause of it, the simplest explanation comes from observing that such an off-set, corresponding to around 27 pixels, corresponds to
a displacement of only 0.2 mm between the lens axis and the centre of the aperture. To form a correction function symmetric





about the zenith, the original curve was "mirrored" about the zenith (see Fig. 8, red curve), then a Gaussian of the form shown in Eq. (5) was fitted to the average of the original and the mirror curve (see Fig. 8, dashed curve).

$$f(z) = A + B \cdot e^{-(z/C)^2} \tag{5}$$

A, B and C so obtained are 0.74, 0.26 and 40.03, respectively. The fitting curve provides a working approximation of the actual DC. Fig. 6 (black-dashed line) shows the SPF corresponding to the raw image from Fig. 3 when geometric, AM, vignetting and mask corrections are included.

## 2.6 Air mass correction

The additional scattering that light undergoes for slant paths is associated with increased image brightness, evident for large $z$. In this context we will be assuming single scattering approximation. To model relative air mass (AM), the ratio of the absolute optical air mass M calculated along $z$ to the zenith air mass $M_0$, a non-refracting homogeneous radially symmetric atmosphere was assumed. This provides realistic values of AM near the horizon, where it is less than 40 as it is expected (Rapp-Arraras and Domingo-Santos, 2011). Each pixel brightness was then corrected by dividing for the corresponding AM($z$). However, a kink in the SPF, at a scattering angle corresponding approximately to the edge of the mask, was observed. This is ascribable to the large value that the AM takes for large $z$, causing the image brightness to drop rapidly. If such values of the AM are excluded from the correction the kink gets significantly reduced. Furthermore for large solar $z$ a cut-off angle that is too large can cause the HR to decrease due to multiple scattering affecting the part of the halo below the sun. It has been shown previously through radiative transfer calculations accounting for multiple scattering that the brightness of the lower part of the halo reaches a maximum for smaller cloud optical thickness $\tau$ than the portion above the sun (Gedzelman and Vollmer, 2008). By setting the cut-off at $z=70°$ the SPF becomes smooth and multiple scattering effects are reduced, while avoiding seasonal bias due to the solar $z$ range covered during the year.

## 2.7 Cirrus discrimination

A quantification of the temporal fluctuations of the infrared brightness temperature BT is expressed in terms of the fluctuation coefficient FC. The FC is obtained here through Detrended Fluctuation Analysis (DFA) of the BT, and is expressed as the exponent of the DFA function expressed as a simple power law (Brocard et al., 2011). It allows detecting the presence of clouds, unless very thin optically. In fact BT fluctuates significantly under optically thick clouds while under clear sky it follows the relatively slow-changing water vapour diurnal cycle. Therefore clear sky FC allows setting a threshold (DFA threshold) for the transition from clear to cloudy sky. This threshold was found empirically to be 0.02 on the basis of the DFA output calculated every 20 minutes and over a time scale ranging between 60 and 150 seconds. The DFA algorithm was applied to one year of data and the corresponding FC time series was averaged over clear sky periods to set the DFA threshold. Separately, a departure from modelled clear sky BT due to the presence of relatively optically thick cirrus can provide a BT threshold (Ci threshold) that was used for assessing cloud phase. A simple analytical model of downwelling thermal radiation under clear skies was





implemented for this purpose. The model uses ground-level air temperature and integrated water vapour path (retrieved locally from GNSS delays) as input parameters to estimate clear sky BT at the central wavelength of the pyrometer (Dandini, 2016). In order to establish via BT whether warm or cold clouds are present in the field of view of our instrument, an estimate of the departure of BT from clear sky BT due to cirrus was calculated. We set the maximum possible departure from clear sky BT

attributable to cirrus as the one when the cloud is warmest and optically thick. Warm liquid clouds would certainly determine a departure from clear sky BT larger than the one due to such cirrus and hence would lend themselves to be discriminated. By assuming an optically thick cirrus (emissivity of 1) at a temperature of -38°C, representative of the upper temperature limit of ice clouds (Heymsfield et al., 2017), an estimate of the irradiance due to the direct emission of cirrus corrected for the atmospheric attenuation, was obtained (see Fig. 10 and 12, middle plot, black dashed line). The Ci threshold follows the

diurnal water vapour cycle corrected for the attenuated contribution of such thick cirrus. Above this threshold we can expect optically thick clouds warmer than -38°C, i.e. theoretically not cirrus. Hence, HR time series corresponding to two test cases discussed in the coming section are complemented by simultaneous comparisons of BT, with the Ci threshold, and the FC with the DFA threshold, as well as broadband downwelling irradiance $I$.

## 3   Results and discussion

We now contrast two case studies based on two consecutive days of observations. Halo ratio time series were obtained on the 6[th] of July 2016 between 8 and 11 am, when halo and non-halo cirrus alternated with scattered cumuli- see Fig. 9 and on the 7[th] of July 2016 between 12 noon and 3 pm, when mostly cirrus occurred - see Fig. 11. All-sky images corresponding to HR minima and maxima are shown as inserts in the figures. Arrows, specifying the time the image was acquired, are black if the halo is either absent or faint, and yellow otherwise. In variance from the previous definitions of the HR (see Introduction), it

was determined as the ratio of the SPF at slightly larger angles: 23° and 20°, which correspond to the locations of the maximum and minimum we typically found in the measured SPF, respectively. This finding corroborates that of Lynch et al. (1985) who observed these values to be 22.8° and 19.7°, respectively. This change resulted in enhanced sensitivity of the HR to the halo status of cirrus - see Fig. 10 and Fig. 12, top plots. The shift of the halo peak towards larger angles than the 22° shown by the more familiar single-scattering SPF computed from geometric optics can be interpreted as originating from the combined

contributions from background sky scattering, diffraction effects (due to small crystal size) and crystal roughness (Macke et al., 1996; Ulanowski et al. 2005; Liu et al., 2013; Smith et al., 2015).

### 3.1   Test case 1: 6[th] of July 2016

On the 6[th] of July between 8 and 10 am cirrus occurs as all-sky camera, BT observations and FC measures all show - see Fig. 10. The FC is nearly always above the DFA threshold while the BT is below the Ci threshold. Over this time window, on

qualitative grounds, the behaviour of the BT and the solar irradiance suggests that the HR increases with the optical thickness $\tau$ (see Fig. 10, middle and bottom plots). This is consistent with the results of simulations (Khokanowsky, 2008), based on ray tracing techniques incorporating physical optics (Mishchenko and Macke, 1998) and neglecting molecular and aerosol



scattering, that show a linear increase of halo contrast with increasing $\tau$ up to $\tau=3$ and a decrease for $\tau>3$ due to multiple scattering. Between about 8 and 8:20 am the HR, mostly <1, shows a maximum and a minimum at about 8:06 and 8:12 am, respectively, when a relatively faint 22° halo is visible. Maxima with HR>1, on the other hand, are measured at 8:33 and 9:06 am, when bright-halo is observed, and at 9:30 am when lower halo brightness is associated with decreased HR. Similarly,

the attenuated halo brightness at about 8:51 am corresponds to an HR minimum. Between 9:36 and 10 am the HR drops below 1 and cirrus gets thinner. The small local maxima observed over the same time window at about 9:38, 9:45 and 9:54 am are ascribable to cirrus optical depth variations. From 10 am, as cirrus disperse, cumuli start entering the field of view of the camera. A local HR maximum is then measured at 10:03 am whereas HR values >1 at 10:16, 10:33 and 11 am, while scattered cumuli over a mostly clear background increasingly occur. The FC testifies to the presence of clouds whereas the BT

becomes larger than the Ci threshold only at about 10:54 am, probably because of the sparse nature of the cumuli preventing significant direct radiation from falling within the field of view of the radiometer. However, dips of the solar irradiance $I$ at approximately 10:24, 10:32 and 10:42 am, indicating increased optical depth, are also observed. While on an overcast day global irradiance depends primarily on diffuse irradiance (Kaskaoutis et al., 2008), on a partly cloudy day clouds crossing the sun path are the main factor to determine variations in the signal of the pyranometer. In particular the ratio of diffuse to global

irradiance has been estimated to be 0.15, 1 and > 0.15 under clear, overcast and partly cloudy sky, respectively (Duchon and O'Malley, 1999; Orsini et al., 2002). These relatively large HR values are artefacts caused by the image becoming brighter at larger scattering angles, possibly due to bright cloud edges. Such cases should be screened out from the analysis as the HR parameter applies to ice clouds only. This example shows that while a relatively large HR (HR>1) can be an indication of the presence of halo-producing cirrus, this does not have always to be the case.

## 3.2   Test case 2: 7th of July 2016

On the 7th of July between 12 noon and 3 pm the sky is mainly characterized by the presence of cirrus, except between about 12 and 12:30 pm when sparse cumuli are seen, around 12:45 pm when relatively opaque altocumuli occur, between around 1:30 and 1:48 pm and about 2:15 pm when cumuli overlap with the cirrus background - see Fig. 11. Correspondingly, the FC is always above the DFA threshold, while the BT is mostly below the Ci threshold although larger values are observed at about

12:20, 12:27, 12:42, 12:48, 1:40, 2:15 and 2:18 pm when, as expected, the solar irradiance $I$ drops. The HR peaks measured at about 12:06 and 12:18 pm, in similarity with the previous case, appear to be associated with bright cloud edges. Between 12:30 and 12:38 pm, when bright-halo is visible, the HR is >1 and then decreases until about 12:42 pm by which time the halo is no longer visible, possibly due to multiple scattering associated with the increased $\tau$, as demonstrated by decreased solar irradiance at that point - Fig. 11. With halo producing cirrus present again from 12:50 pm, the HR increases, except for a local

minimum at about 1 pm, until about 1:18 pm, when it becomes larger than 1 and the halo is correspondingly sharper. The HR then drops fairly steadily, except for a local maximum at about 1:30 pm, while the halo, still partly visible, fades gradually away to eventually disappear by about 1:36 pm. This is when, we hypothesize, cloud $\tau$ has become <1 and Rayleigh and aerosol scattering contribute to the halo contrast reduction. A faint halo present from about 1:45 pm gets quite sharp by 2 pm as the HR becomes again larger than 1. The halo brightness then stays constant for about ten minutes before the HR reaches another



minimum at about 2:12 pm, when cumuli are seen by the camera. As the halo sharpness increases, the HR also increases fairly uniformly until about 2:28 pm, when a maximum of 1.14 is reached. We speculate, based on previous findings (Forster et al., 2017; Kokhanowsky, 2008), that this is when $\tau$, probably somewhere between 1 and 3, becomes significantly larger than Rayleigh and aerosol optical thickness without exceeding the optical depth beyond which multiple scattering becomes

dominant. The HR then drops to 1 in less than 20 minutes, while the halo is still fairly bright, and continues to go down until 3 pm when the halo is eventually no longer visible. Peaks at about 2:40 and 2:54 pm are due to rapid variations in sky brightness, associated with cumuli like those observed at near 2:38 pm, which, in analogy with the minima seen at 2:12 and 1:48 pm, cause the SPF to vary significantly as the cumuli transit over the cirrus. With an average BT roughly 10°C higher than on the previous day, this is a case of relatively warm halo-producing cloud, yet the relatively large HR indicates that the cloud is dominated by

ice.

Overall the HR correlates well with the fluctuating halo visibility observed throughout the periods examined. According to previous radiative transfer calculations (Forster et al., 2017), we conjecture that in order to observe the HR maxima measured for the two cases discussed here a certain minimum fraction of smooth hexagonal ice columns had to be present. If multiple scattering were to be accounted for, such a fraction could be an underestimation of the actual one.

**4   Conclusions**

A method for the retrieval of the halo ratio HR from all-sky imaging has been proposed. This consists of applying a series of image transformations and corrections needed to interpret images quantitatively in terms of the scattering phase function. Halo formation can then be identified by taking a ratio of phase function values at particular scattering angles in the vicinity of the halo peak. Unlike in previous studies which tended to use slightly smaller angles (Auriol et al., 2001; Gayet et al.,

2011; Ulanowski et al., 2014; Forster et al., 2017) we have used a ratio at scattering angles of 23° and 20° corresponding to the locations of the maximum and minimum we typically found in the measured SPF, respectively. The new angles result in higher values of the HR from our data than the HR definitions cited above. After applying the corrections and transformations, HR time series have been shown for two test cases, the 6[th] and 7[th] of July 2016. HR values $\geq$1 were observed under halo producing cirrus, but also sometimes under scattered low-level clouds when HR maxima appeared to be artefacts due to bright

cloud edges. As previously predicted (Kokhanovsky, 2008) multiple scattering appears to lead to decreased HR. We have partly counteracted this by excluding from the HR calculations pixels at $z$>70°. However, in future it would be possible to exclude the lower parts of the halo where the slant optical thickness is too large. This implies having to estimate the cirrus optical thickness $\tau$, which could be derived from pyranometer measurements, for example (Fitzpatrick et al., 2005; Qiu, 2006) - allowing, by the way, to verify the expected relation between HR and $\tau$ which sets a maximum HR at $\tau$=1 (Forster et al., 2017). Overall

the HR is shown to be sensitive to the halo status of cirrus as it is well correlated with halo visibility. All-sky cameras have the advantage of being relatively cheap when compared to more complex and difficult to align tracking systems such as the HaloCam (Forster et al., 2017). Moreover like the sun-tracking camera used by Foster et al. (2017), who have quantified the shift of the red tinged inner edge of the 22° halo, colour all-sky cameras can also provide spectral dependence.



However, the all-sky camera data should also be supplemented with additional information if the HR observations are to be associated only with cirrus: a separate cirrus detection method is necessary to screen out non-ice clouds and clear sky periods, before deriving reliable HR statistics. The quantification of the fluctuations of the brightness temperature BT, expressed in terms of the fluctuation coefficient FC, has been used to discriminate clouds from clear sky by comparing the FC to the

DFA threshold which is used to set the transition from clear to cloudy skies. However, for very small $\tau$ the fluctuations of the BT can be of similar magnitude as under clear sky, putting a limit on this technique in the context of very thin cirrus. Additionally, an estimate of the magnitude of the BT in the presence of optically thick cirrus has been used as an indicator of cloud phase. This method has managed to detect cirrus most of the time over the periods of observation. However it was unable to discriminate some of the scattered cumuli, sometimes associated with high HR values that have to be screened out from

the analysis, and in one case failed to confirm the presence of ice, as shown by the presence of the 22° halo. Where no other sky observations are available and attenuation of solar irradiance due to aerosol can be measured or accounted for, a method previously implemented for solar irradiance time series from pyranometer measurements can be used for cloud classification (Duchon and O'Malley, 1999). This can be achieved by accounting for the standard deviation of the scaled observed irradiance and the ratio of the former to the scaled clear sky irradiance (Duchon and O'Malley, 1999) or, as a cheap and relatively easy

to use alternative, by combining observed total irradiance, temperature and relative humidity (Pagès et al., 2003). In the future these cirrus discrimination methods should be compared to techniques such as microwave radiometry or lidar which would allow us to assess their relative merit. As an alternative to BT radiometry the backscattered signal from lidar can be used to discriminate between cloudy and clear skies. If depolarization information is not available cloud phase discrimination can be achieved from cloud base height by estimating cloud base temperature. Such estimates can be improved if temperature profiles

are available from radiosonde ascents (Forster et al., 2017). While this brings the advantage that the cloud base temperature will be less sensitive to $\tau$ than the temperature from the radiometer on the other hand this method requires additional measurements. However it can be considered an alternative if such measurements are available at the given observation site.

Ultimately, the method proposed here is meant to provide cirrus characterization. The two test cases analysed show the presence of large (compared to wavelength, probably characterized by size parameters larger than 100 - Mishchenko and

Macke, 1998) and regular, smooth ice crystals. Results of previous investigations (Forster et al., 2017) have been used to speculate on such smooth crystal fraction. We argue that when the 22° halo was visible a percentage of at least 20% of regular ice crystals had to be present if molecular, aerosol and multiple scattering in addition to surface albedo are accounted for, together contributing to a reduction of the halo contrast. We have also conjectured that when the HR reached its absolute maxima the fraction of such pristine crystals is likely to have been much larger than 20%. The remaining fraction could have

been composed of irregularly shaped, complex, rough or small ice crystals.

Long term observations of halo displays, preferably at multiple sites, must be carried out to allow obtaining statistics of the occurrence of halo producing cirrus, which still remains unknown. The magnitude of the HR could then be used to assess aspects of the composition of the cirrus - while remembering that low HR can have multiple causes, as discussed above. Furthermore, by extending the method to additional halo displays further information on ice crystal geometry could potentially





be obtained - e.g. the presence of sun dogs and the 46° halo indicates the presence of aligned plates and solid columns, respectively.

The utilisation of the all-sky cameras to transform the measured light intensity into the scattering phase function and, on a limited extent, the cirrus detection algorithm, are the particularly novel aspects of this work; this has not been done previously to

5   the best of our knowledge. The method applied to the all-sky images in particular, allowing the measurement of the distribution of sky radiance, permits taking advantage of the large field of view associated with the all-sky imaging in a quantitative manner. Consequently, while not computationally demanding and relatively easy to implement, this method allows broadening the range of application of the all-sky imaging beyond the more qualitative recording of cloud fields and optical displays associated with cirrus. The cloud classification method, on the other hand, is original in that relies on a non-fixed, non-location specific and

10   easy to model temperature threshold. The combined use of these two methods, allows relatively inexpensive halo observations and the retrieval of information pertaining to ice particles size and texture. If implemented at multiple locations, the methods can provide a useful dataset for improving the understanding of cirrus composition.

*Competing interests.*   No competing interests are present.

*Acknowledgements.*   The services of the Natural Environment Research Council (NERC) British Isles continuous GNSS Facility (BIGF),

15   www.bigf.ac.uk, in providing archived GNSS products to this study, are gratefully acknowledged. We thank Prof. Paul Kaye for his contri-
bution to designing the occulting disc, Dr. Evelyn Hesse (University of Hertfordshire) and Dr. Anthony Baran (Met Office) for their valuable
advice.





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





**Figure 1.** All-sky daytime camera with occulting disk in operation in the foreground. The night time camera is in the background and the right.





**Figure 2.** Predicted (blue dots, from Eq. (1) and (2)) and actual (red dots) star and planet trajectories. Each trajectory is labelled with the name of the star-planet and the mean angular difference between predicted and actual positions.





**Figure 3.** All-sky daytime camera image obtained on the 10th of April 2016 at 10:26 am showing horizon (red solid circle).



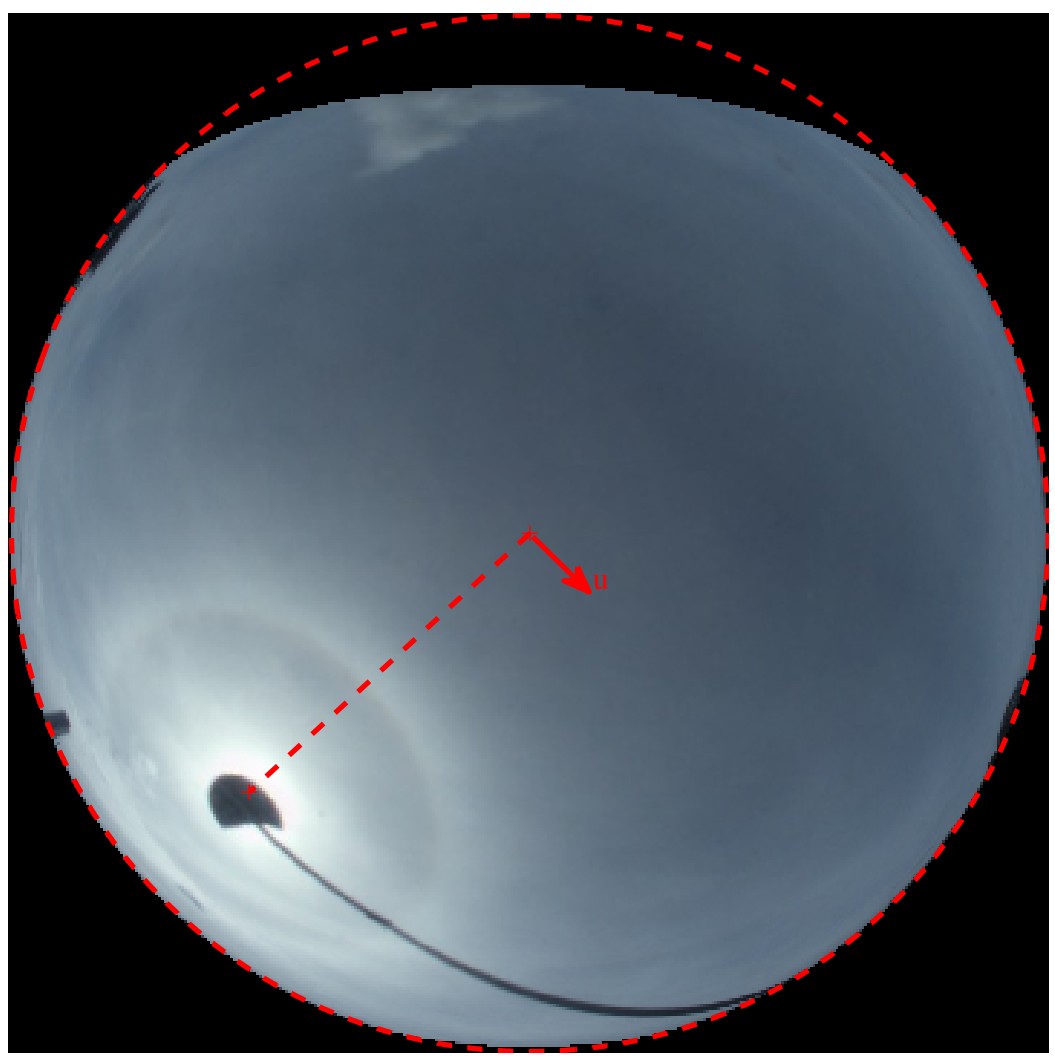

**Figure 4.** Projection of the original image from Fig. 3 onto a sphere (upper hemisphere).



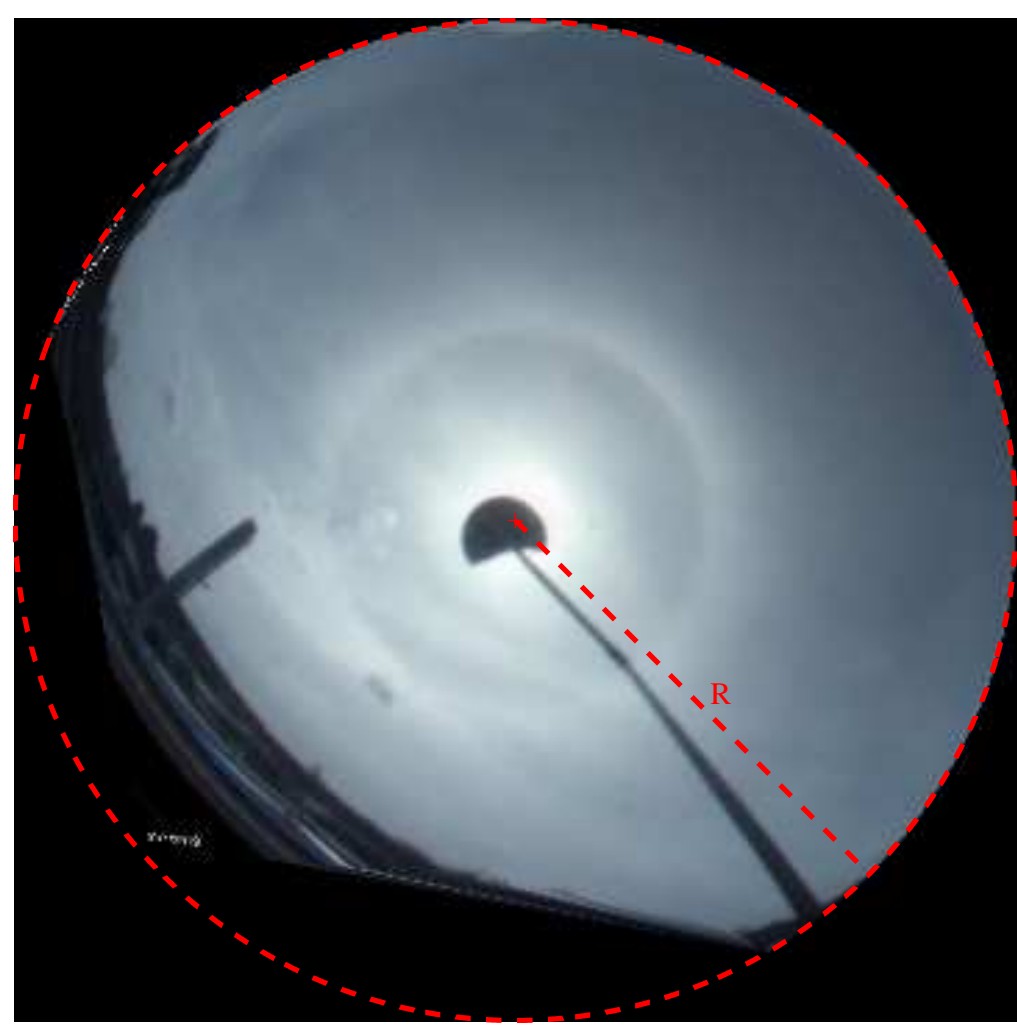

**Figure 5.** Image from Fig. 4 after rotation (upper hemisphere only).





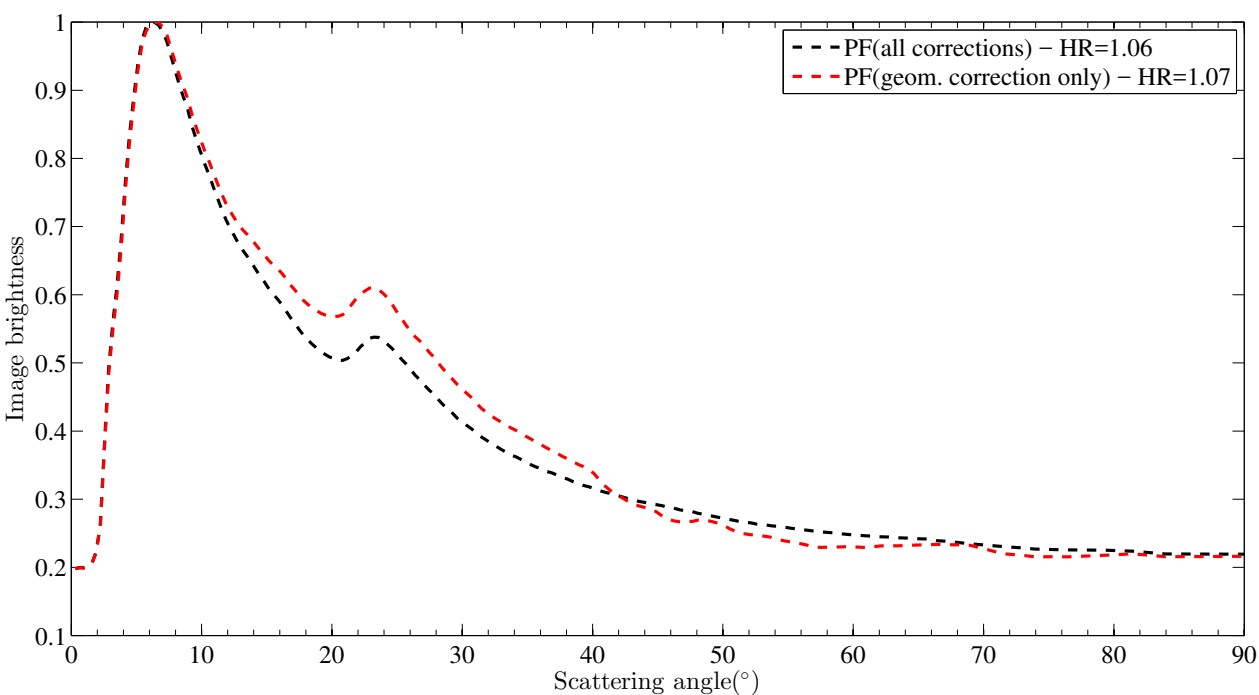

**Figure 6.** Measured scattering phase function corresponding to image in Fig. 3 with geometric correction only - red-dashed, with geometric, air mass, mask and vignetting corrections - black-dashed. The corresponding HR measures are also shown.



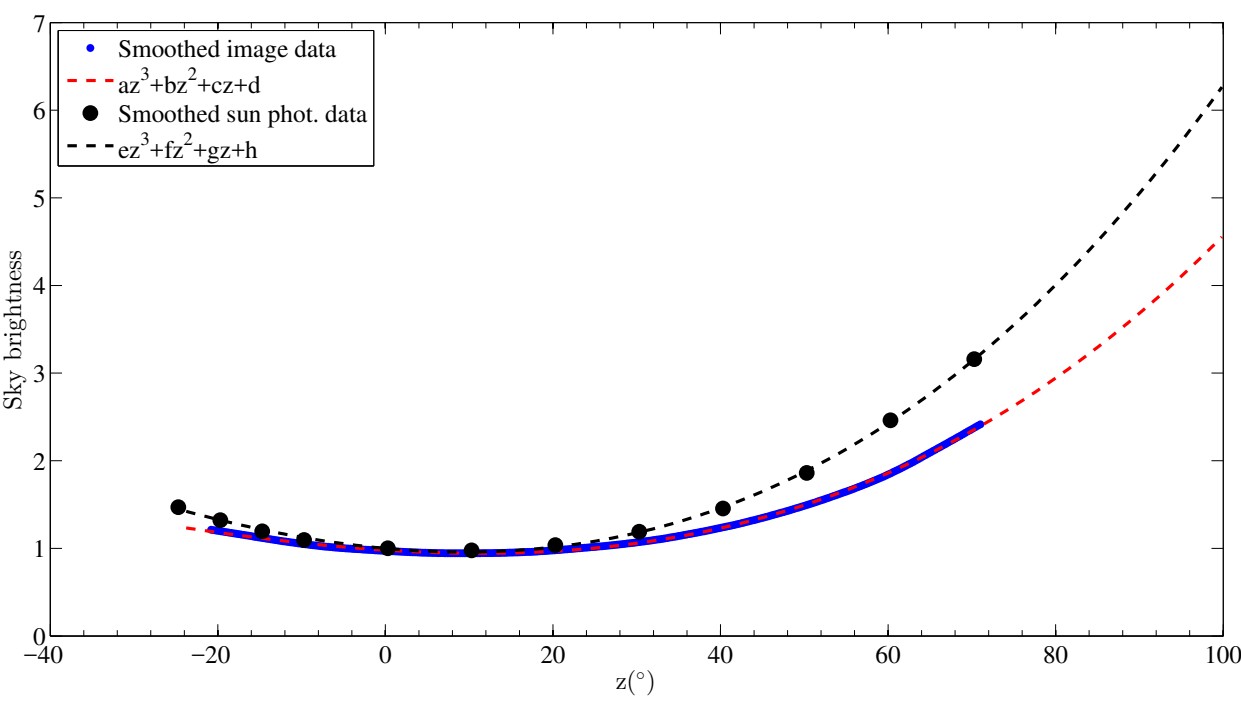

**Figure 7.** All-sky camera and sun photometer normalized fitting polynomials (black-dashed line - Sun photometer, red-dashed line - Camera).



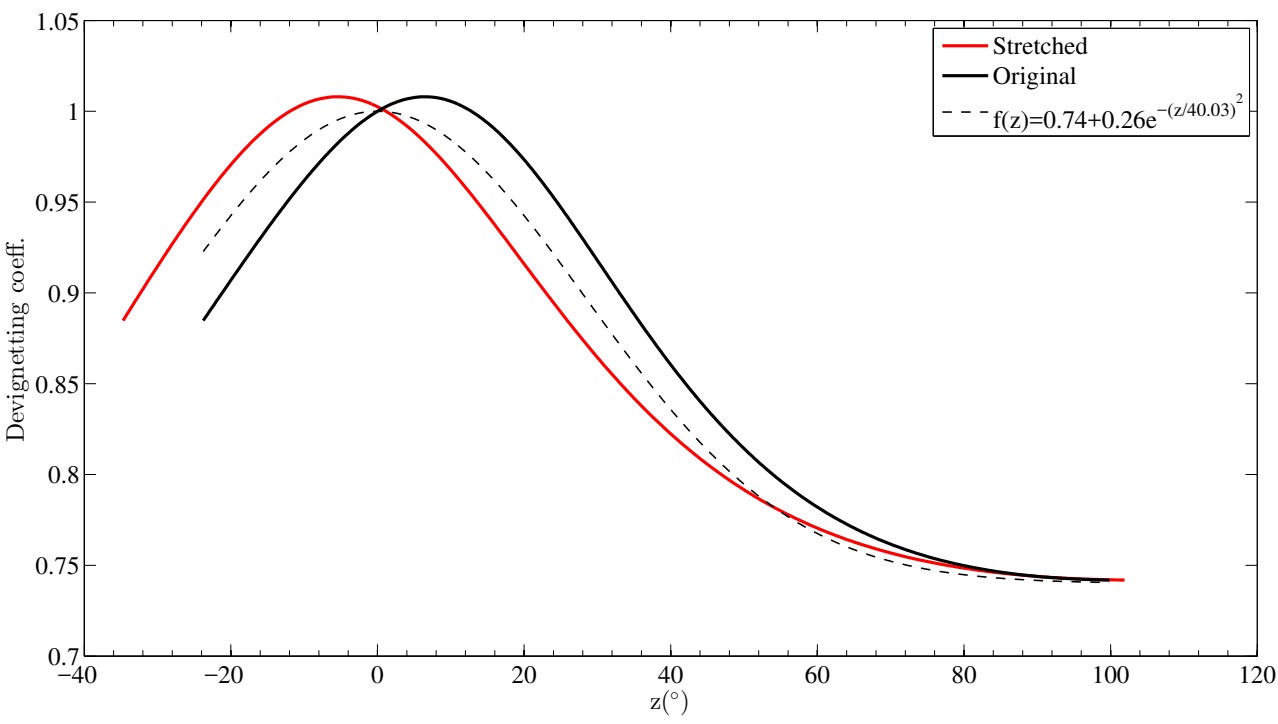

**Figure 8.** Devignetting coefficient: black line: original data, red line: stretched curve, dashed line: fit.



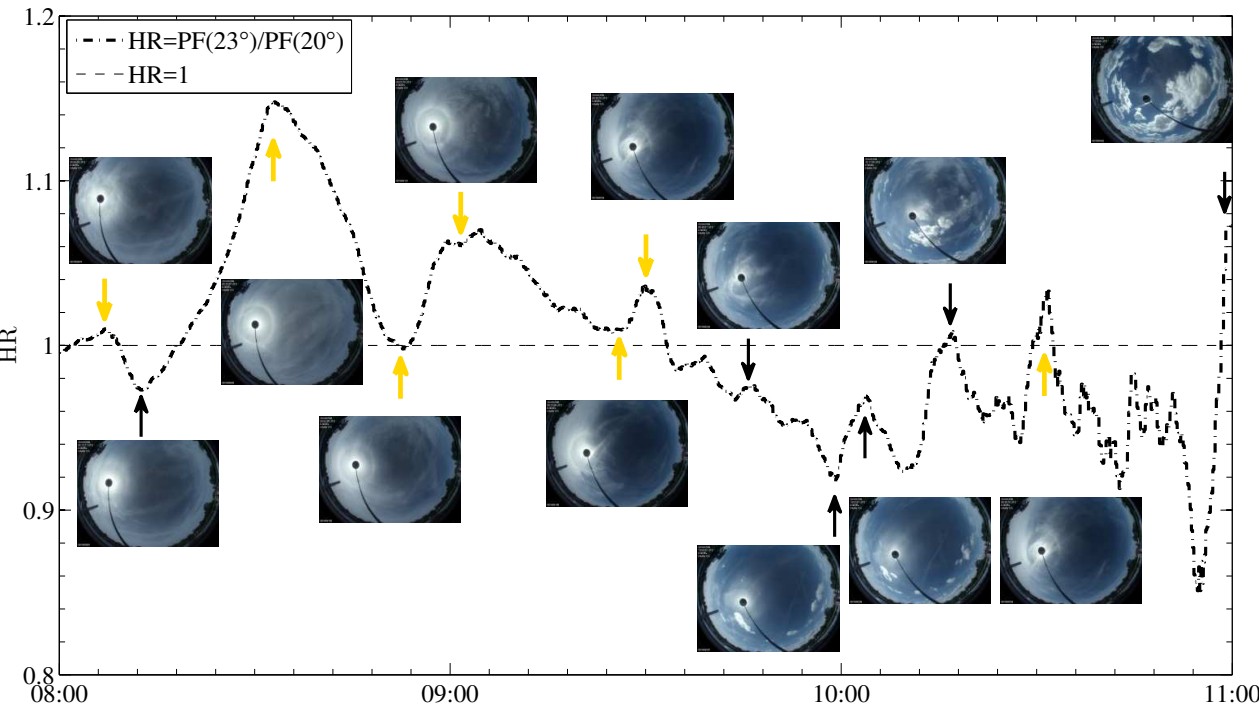

**Figure 9.** HR time series from 6$^{th}$ July 2016 between 8 am and 11 am. All-sky images corresponding to peaks and dips are also shown. Yellow arrows indicate presence of relatively bright halo while black ones indicate either presence of faint halo or absence of it.




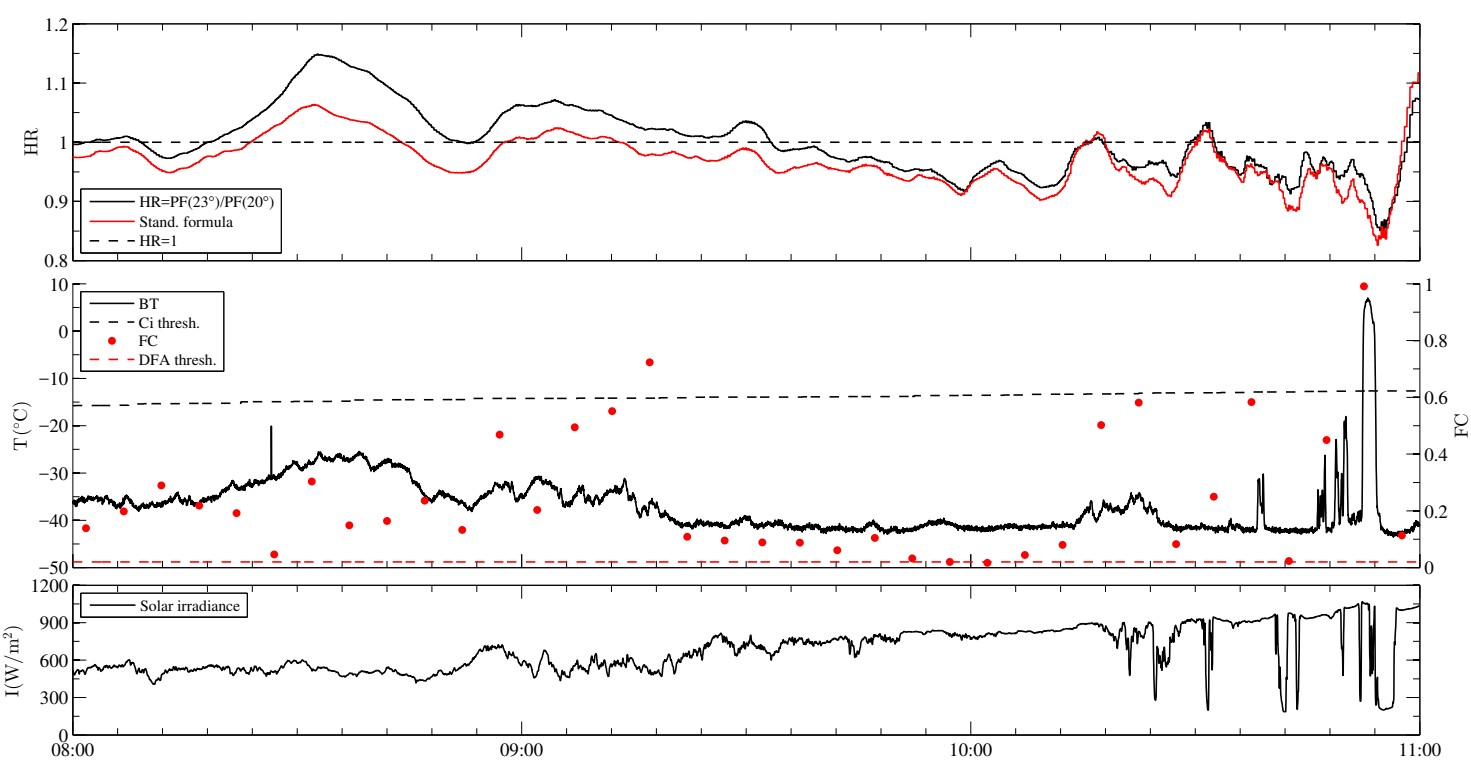

**Figure 10.** Time series from 6th July 2016 between 8 am and 11 am. Top plot: HR, black line - new definition, red line - standard formula - (see text); middle plot: BT - black line, FC - red dot, DFA threshold - red-dashed line, Ci threshold - black-dashed line; bottom plot: solar irradiance.





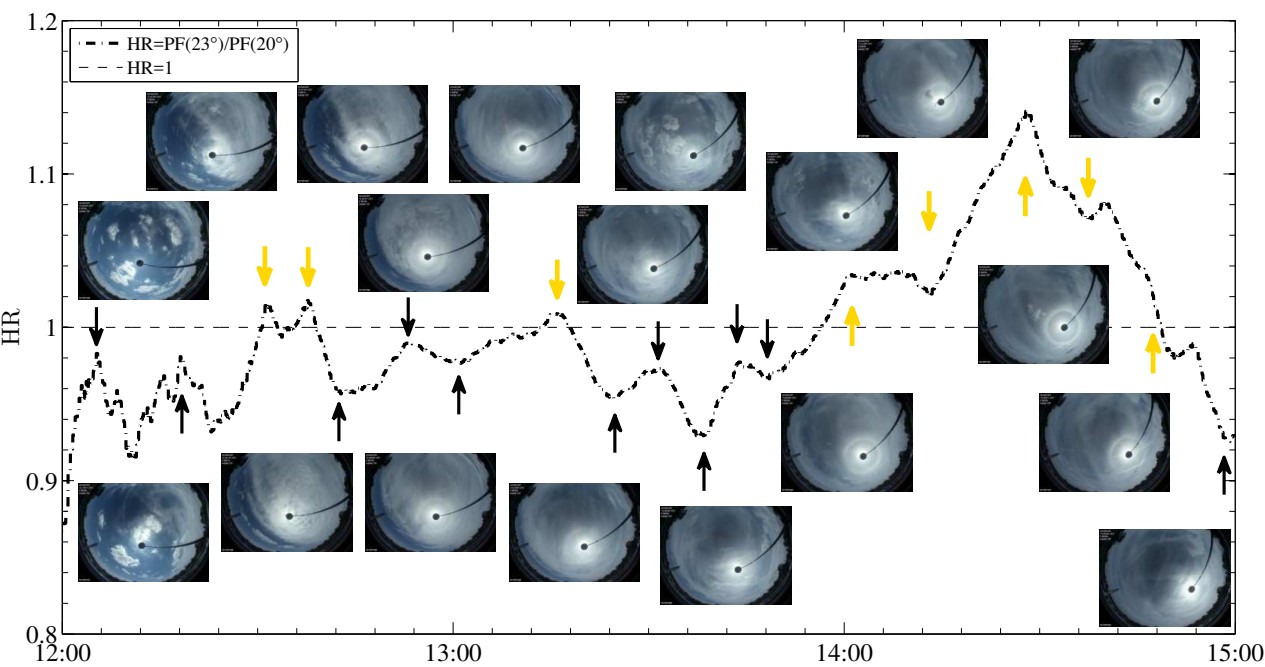

**Figure 11.** As Fig. 9, but HR time series from 7[th] July 2016 between 12 noon and 3pm.





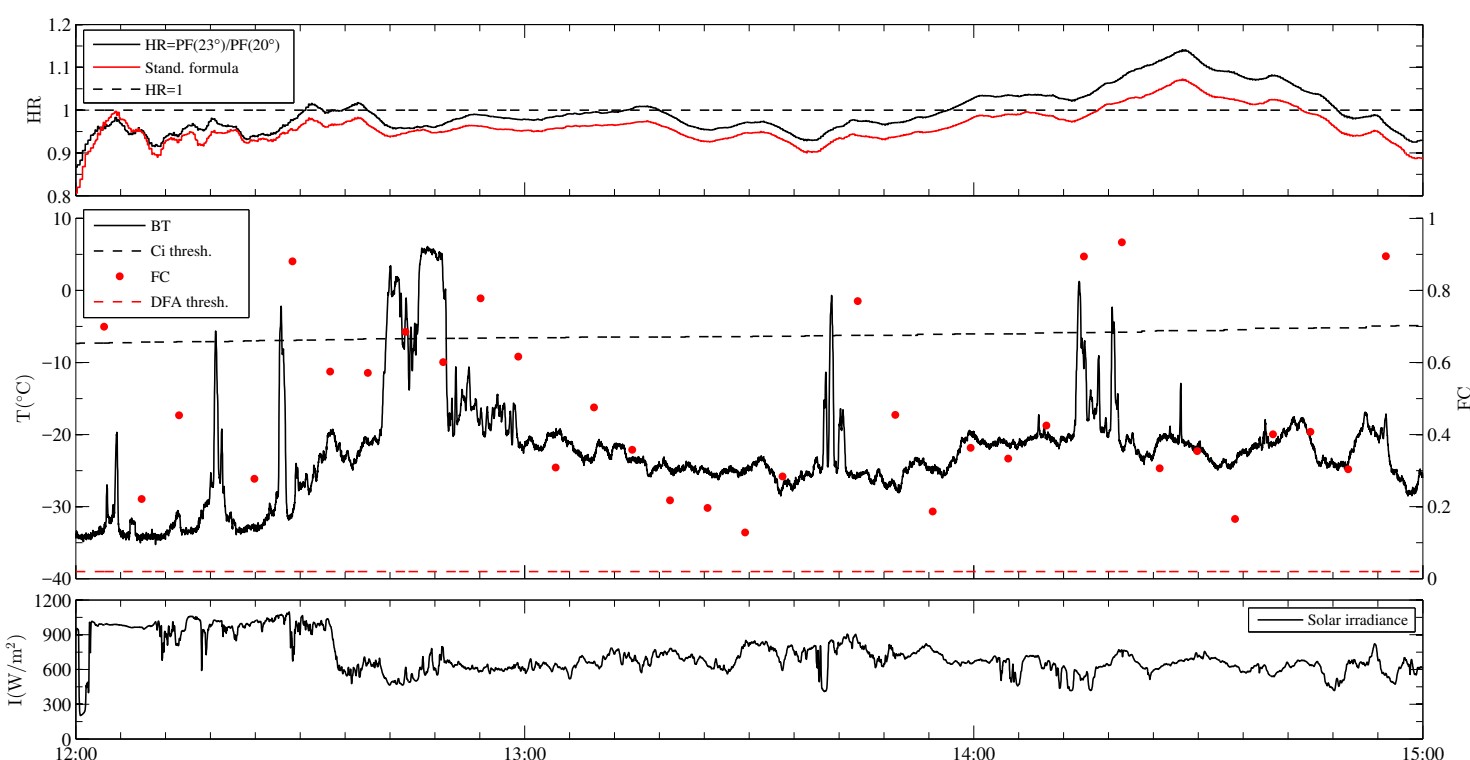

**Figure 12.** Time series from 7[th] July 2016 between 12 noon and 3 pm. Top plot: HR, black line - new definition, red line - standard formula - (see text); middle plot: BT - black line, FC - red dot, DFA threshold - red-dashed line, Ci threshold - black-dashed line; bottom plot: solar irradiance.