# Peer review of "Halo ratio from ground based all-sky imaging"

_Atmospheric Measurement Techniques, 2018_

## Referee Comment (RC1) · Anonymous Referee #1 · 7 Mar 2018

The manuscript presents an interesting approach to evaluate the halo ratio from all-sky imaging and using infrared brightness temperature measurements to identify cirrus clouds. However, the method description lacks some information and is partly not coherent. Moreover, several conclusions drawn from the presented case studies are not supported by the observations. Also some misconceptions occurred:

- Abstract, Page 2, line 32 and especially Page 6, line 20:
  The term "scattering phase function" should not be used in this context. It seems that the analysis in this study is based on the measured brightness distribution of the camera which includes multiple scattering rather than the actual scattering phase function, which is a single scattering property.

- Page 3, line 9:
  "...the reflectivity is inversely proportional to the HR." and line 14: "The asymmetry parameter [...] is expected to be positively correlated with HR"
  These statements are based on observations (Ulanowski et al., 2006; Gayet et al., 2011; Ulanowski et al., 2014) with most HR values < 1, i.e. no 22° halo, which has to be emphasized in this context. Since it is questionable whether differentiating values HR<1 is meaningful at all, the relationship between HR and asymmetry parameter should be explained here first without assumptions:

  In general, the reflectivity is primarily determined by the asymmetry parameter which depends on the ice crystal surface roughness and aspect ratio in a U-shaped manner. The HR of the scattering phase function increases for ARs ranging from plates over compact crystals to columns. This implies an ambiguity in the relationship between HR and asymmetry parameter and thus also on the reflectivity.
  See
  ○ van Diedenhoven 2012: Remote sensing of ice crystal asymmetry parameter using multi-directional polarization measurements – Part 1: Methodology and evaluation with simulated measurements
  ○ van Diedenhoven 2014: The prevalence of the 22° halo in cirrus clouds
  ○ van Diedenhoven 2014a: A Flexible Parameterization for Shortwave Optical Properties of Ice Crystals

- Many abbreviations and acronyms, especially in Sections 2.7 and 3, make the paper difficult to read.

In the following more detailed comments:

1. **Introduction:**
   The Introduction mainly focuses on cloud detection using all-sky imagers. Are these methods relevant for the presented study?

2. **Section 2.2.1:**
   ○ Please provide some references for the calibration method using the coordinates of stars.
   ○ How does the fit to a bi-cubic function work? Please provide the complete camera model used for the calibration.
   ○ Which kind of distortion is corrected – tangential or radial or both?
   ○ How many pictures are used for the calibration?

- What is the effect of increasing the lens aperture on the accuracy of the geometric calibration?

3. **Section 2.2.2:** The authors state a reprojection error of "mostly <0.1° aside from portions of […] trajectories for which larger discrepancies, up to 0.38°, were observed" . How large is the error in the region of interest, i.e. where the 22° halo occurs? Is the accuracy sufficient for this study?

4. **Section 2.5:** Using sun photometer measurements to estimate the vignetting effect of the camera is an interesting approach, but some issues are not discussed.
   - Page 7, line 20: What is the spectral response of the blue channel? The influence of Rayleigh scattering in the blue channel is much stronger compared to the red channel and small deviations in the considered wavelengths might cause larger errors in the radiance distribution. Why not use all 3 (RGB) channels? What is the required accuracy of the vignetting correction?
   - Page 7, line 23: "...by neglecting pixels such that z is greater than approximately 20° over the meridian containing the sun...". Why are these pixels neglected?
   - Page 7, line 27: "...the presence of a peak located roughly 8° from the zenith was investigated...To form a correction function symmetric about the zenith, the original curve was "mirrored" about the zenith.." Based on which assumption? Is this peak is related to the camera or the sun photometer data? Was this peak observed at a different time as well?
   - Are the measurements interpolated to the same time? How long did the sun photometer scan take?
   - Is the vignetting correction determined in the principal plane applied to the whole camera image? If yes, under which assumption?
   - Fig. 6: Why is the calibrated brightness distribution (black dashed line) smoothed compared to the original data (red dashed line)?

5. **Section 2.6:** What is the purpose of the air mass correction? Is the applied method of Rapp-Arraras and Domingo-Santos (2011) applicable to cloudy scenes as well? What is the error in this case?

6. **Section 2.7:**
   - The cirrus BT threshold is based on an optically thick cirrus. 22° halos are only visible in thin cirrus. What is the effect of a decreasing optical thickness on the BT threshold?
   - How sensitive is the presented threshold method on variations of cloud cover in the scene?
   - The DFA threshold was estimated empirically to 0.02. This value is much lower than the values found by Brocard et al. 2011 (~0.1 for clear sky, ~0.5 for stratiform cirrus, ~1 for broken cirrus), why?

7. **Section 3.1:**
   - Page 9, line 28: "...between 8 and 10 am… Over this time window […] the behaviour of the BT and the solar irradiance suggests that the HR increases with the optical thickness τ (see Fig. 10, middle and bottom plots)" This is not clearly visible in Fig. 10. The HR increases from 8:12 until 8:30 am but clearly decreases towards 10 am. How is the optical thickness derived from the displayed data?
   - Page 9, line 31: Kokhanovsky 2008 showed that the halo contrast is linearly decreasing with increasing optical thickness since molecular and aerosol scattering are neglected. Only **the radiance distribution is increasing up to τ=3 and decreasing for τ>3.**

- Page 10, line 2: "Between about 8 and 8:20 am the HR, mostly <1, shows a maximum and a minimum at about 8:06 and 8:12 am, respectively, when a relatively faint 22° halo is visible"
  How can the HR be <1 but the image shows a 22° halo? The 22° halo in the image at 8:54 appears brighter than at 8:06 but the HR is smaller. Why? Is this an effect of averaging over the complete 22° halo scattering angle region? If so, the HR might be primarily a measure of cloud fraction in this case.

8. **Section 3.2:**
   Page 11, line 11: "... we conjecture that in order to observe the HR maxima measured for the two cases discussed here a certain minimum fraction of smooth hexagonal ice columns had to be present. " The second sentence is now redundant.

9. **Conclusions:**
   - Page 11, line 29/30: " Overall the HR is shown to be sensitive to the halo status of cirrus as it is well correlated with halo visibility." Can you provide a quantitative statement?

   - Page 12, line 3-9: Discrimination between cloud and clear sky seems to work well using the halo ratio alone, without additional DFA threshold test. Discrimination between cirrus and broken cumulus or other clouds seems necessary. How (well) does it work?

   - How does the presented method perform compared to the methods presented in the Introduction for cloud type discrimination and halo detection? (e.g. Shields et al. 2013, Forster et al. 2017)

   - Page 12, line 26: "We argue that when the 22° halo was visible a percentage of at least 20% of regular ice crystals had to be present if molecular, aerosol and multiple scattering in addition to surface albedo are accounted for, together contributing to a reduction of the halo contrast." What is the basis of this argument? Please explain.

   - Page 12, line 28: "We have also conjectured that when the HR reached its absolute maxima the fraction of such pristine crystals is likely to have been much larger than 20%." See previous comment.
     "The remaining fraction could have been composed of irregularly shaped, complex, rough or small ice crystals." In Sections 3.1 and 3.2 fluctuations of the HR are explained only by variations of the cloud optical thickness, now the argument is based on the ice crystal microphysical properties. This is not consistent.

   - Page 13, line 1: should be "...e.g. the presence of sun dogs and the 46° halo indicates the presence of aligned plates and **randomly oriented hexagonal crystals**, respectively."

   - Page 13, line 3: "The utilisation of the all-sky cameras to transform the measured light intensity into the scattering phase function and, on a limited extent, the cirrus detection algorithm, are the particularly novel aspects of this work; this has not been done previously to the best of our knowledge." The term "cirrus detection algorithm" appears only in the Abstract and Conclusions. Where is this algorithm explained? The method described in Section 2.7 is not new and was already published by Brocard et al., 2011.

   - Page 13, line 10: " The combined use of these two methods, allows relatively inexpensive halo observations and the retrieval of information pertaining to ice particles size and texture." How can particle size an texture be retrieved?

---

## Referee Comment (RC2) · Anonymous Referee #2 · 7 May 2018

The manuscript describes an interesting approach to automated halo observations using an all-sky camera. Even more important, it includes an automated processing of the data to derive a halo ratio which could be used in further work to retrieve information about shape and size of ice crystals. There is a number of points, however, which need to be clarified before publication.

Major points:

- The term "scattering phase function" is misleading, as I already mentioned in the quick review. The scattering phase function a clearly defined single scattering property, in particular the probability of scattering into a certain direction. What the authors determine here is a normalised radiance distribution. In case of optically very thin clouds (tau«1, single scattering), the radiance distribution would be close to the scattering phase function if there was no additional molecular scattering. Please avoid the

term because it is misleading!

- After the two case studies, I am not sure if the brightness temperature fluctuation threshold provides any useful information. Nearly all shown data points are clearly above the threshold.It would be good to see examples where the values are actually below the threshold.

- Conclusions, page 12, line 26: Where do the numbers (20%) concerning smooth crystal fractions come from? I do not see any evidence in the manuscript, neither any reference. If there is no quantitative evidence, please remove the statement.

- How was the halo ratio determined from the images? Maybe I missed it, but it isn't clear to me if the average over the full circle around the transformed image center was taken, or if only a sector was used, or ... And how does the (admittedly small) angular error affect the halo ratio?

Minor points:

Page 2, line 20: This is not really a sentence and the level of detail could be reduced, since it is anyway not sufficient to understand the technique without having to read the referenced paper.

Page 3, line 23: "previous" findings are contrasted to more recent results. However, these two papers describe observations in completely different locations on the globe which might be one cause for differences.

Page 4, line 3: Since the reader might now be familiar with the details of image vignetting at this point in the text, this section might be moved after the "all-sky-camera" section.

Page 4, line 16: What are these "2.52%"? Fraction of the solid angle? Please define.

Page 4, line 21: I am not sure how many readers are still familiar with kbaud. One might use kbit/s instead.

Page 4, line 25: The shadow disk for the daytime camera is meant to reduce stray light. Isn't the same necessary for the night time camera? The moon is certainly six orders of magnitude darker than the sun, but the lunar halo is as well. Or in other words, the ratio of the brightness lunar halo / moon should be the same as the ratio solar halo / sun. The straylight problem should be the same during day and night.

Page 5, line 27: Could you provide the formula of the bi-cubic function and the order of magnitude of the deviation from the ideal camera model? From statements in the following section I infer that it was actually not used? If so, why mention it at all?

Page 6, line 8: I assume, the test image here was different from the one in the previous section which was used for the geometric calibration?

Page 7, line 16:Could you briefly (one sentence) explain what vignetting is and how it is caused?

Page 7, line 24: Explain acronym LOWESS

Page 7, line 27: with "the presence of a peak located roughly 8°" you mean that the maximum of the function is at 8° rather than at 0° where you would expect it?

Page 8, line 1: Could you please explain why the average between the original and the mirrored curve was used, rather than shifting the curve by 8°?

Figure 8, caption: This is confusing and contradicts the text: The caption says "stretched", while the curve is obviously shifted. The text on the other hand says "mirrored" rather than shifted.

Page 8, line 8: The purpose of the airmass correction is not clear to me. Is it to make the brightness distribution more "flat"?

Page 8, line 19: To assume single scattering the slant optical thickness has to be smaller than 1. For a viewing zenith angle of 70° the slant optical thickness would be about three times the vertical optical thickness, and thus the assumption of single

scattering would only be true for clouds with vertical optical thickness smaller than 0.3. I am sure many of the observed cirrus clouds had a larger optical thickness. Also, the vertically integrated Rayleigh optical thickness in the blue is about 0.2 which adds to the cirrus optical thickness. This once more confirms that you don't observe the scattering phase function but the sky radiance which is caused by cloud scattering plus molecular scattering.

Page 8, line 27: How was the threshold determined empirically? From "manually" deciding if an image is cloud/cloudless and looking at the corresponding fluctuation?

Page 9, line 8: With "irradiance due to the direct emission of cirrus corrected for the atmospheric attenuation" you actually mean after correction to brightness temperature assuming Stephan-Bolthmann? The large deviation from -38°C results from the fact that your broadband instrument integrates over water vapor absorption as well as the atmospheric window? At first glance I expected it to be closer to -38°C, but that would only be the case for a narrowband instrument measuring in the atmospheric window.

---

## Author Comment (AC1) · 31 May 2018

"**Interactive comment on "***Halo ratio from ground based all-sky imaging***" by Paolo Dandini et al.**

"**Response to Anonymous Referee #2**

We thank the Referee for many insightful comments, questions and helpful suggestions that allowed us to improve the manuscript. We list them below, together with our clarifications and changes to the manuscript.

Major points:

- The term "scattering phase function" is misleading, as I already mentioned in the quick review. The scattering phase function a clearly defined single scattering property, in particular the probability of scattering into a certain direction. What the authors deter-

mine here is a normalised radiance distribution. In case of optically very thin clouds (tau<<1, single scattering), the radiance distribution would be close to the scattering phase function if there was no additional molecular scattering. Please avoid the term because it is misleading!

Authors: We do not claim that what we observe is due solely to cirrus, and a phase function remains a phase function even if it is due to a mix of scattering components. It is appropriate to acknowledge that in some cases the observations depart from the single scattering assumption. However, a term like "sky radiance" is not suitable. This is because our results focus on radiance that is corrected for the relative airmass contribution, so is not simply sky brightness. Our use of the term phase function is intentional: the corrections applied to the measured radiance are intended to provide an approximation to the angular dependence of the unnormalized (1,1) element of the scattering matrix. We use the term phase function as shorthand for this property. There is no other term in common use that we are aware of that describes the angular dependence of scattering in this context. We define the meaning of our usage of the term phase function early in the text.

However, for greater clarity, after "*Nevertheless, sky imaging is effective for recording the optical displays sometimes associated with cirrus called halos and for measuring the angular distribution of scattered light.*" we now continue as follows "*We will henceforth refer to this quantity Scattering Phase Function as the corrections applied to the measured radiance are intended to provide an approximation to the angular dependence of the unnormalized (1,1) element of the scattering matrix.*".

- After the two case studies, I am not sure if the brightness temperature fluctuation threshold provides any useful information. Nearly all shown data points are clearly above the threshold. It would be good to see examples where the values are actually below the threshold.

Authors: Cirrus discrimination is only a subsidiary topic of the present article, the main aim being the retrieval and the behaviour of the halo ratio. Therefore, for brevity, we focus almost exclusively on cloudy skies. Brightness temperature fluctuation as a means

to detect cirrus has been the subject of the previous studies.

- Conclusions, page 12, line 26: Where do the numbers (20%) concerning smooth crystal fractions come from? I do not see any evidence in the manuscript, neither any reference. If there is no quantitative evidence, please remove the statement.
Authors: This is based on sensitivity studies of light scattering simulations (Forster et. al., 2017) showing (see Fig. 11 of Forster et. al.) that HR>1 is associated to smooth crystal fraction larger than 20%, as already cited (end of section 3).

- How was the halo ratio determined from the images? Maybe I missed it, but it isn't clear to me if the average over the full circle around the transformed image center was taken, or if only a sector was used, or ... And how does the (admittedly small) angular error affect the halo ratio?
Authors: The halo ratio is determined from the phase function, as already described in the introduction and then extended at the beginning of section 3 with the definition of scattering angles in use. So the Reviewer's question boils down to how the phase function is obtained; this is described at the beginning of section 2.3, as follows: "*The SPF is obtained by averaging the image brightness over pixels which are equidistant from the light source in terms of scattering angle*", and then reinforced at the end of this section with reference to the example image as "*The interpolated image is ultimately used to calculate the brightness as a function of the scattering angle*". For greater clarity we now insert at the end the words "*by averaging over the entire azimuth angle range.*".
As to the effect of the angular error on the halo ratio, we have not studied this aspect, beyond determining that the angular error is small, as the reviewer states.

Minor points:

Page 2, line 20: This is not really a sentence and the level of detail could be reduced, since it is anyway not sufficient to understand the technique without having to read the referenced paper.

Authors: The sentence

"*Field images, a red, a blue and a red and a blue trimmed with neural density were calibrated for the non-linearity of the basic sensor and for differences in the pass bands of the spectral filters and then used to obtain the corresponding blue to red ratio images before a threshold was set for determining the presence of thin clouds.*" has been as modified as follows: "*Field images were used to obtain the corresponding blue to red ratio images before a threshold was set for determining the presence of thin clouds.*".

Page 3, line 23: "previous" findings are contrasted to more recent results. However, these two papers describe observations in completely different locations on the globe which might be one cause for differences.
Authors: While these observations are from different geographic locations they both have been recorded in the northern-hemisphere for a mid-latitude scenario. Here we are not trying to give reasons as to why different studies show different results. We only want to stress that differences exist and should be resolved. Here we propose enabling this by the introduction of an inexpensive and easy to implement technique.

Page 4, line 3: Since the reader might now be familiar with the details of image vignetting at this point in the text, this section might be moved after the "all-sky-camera" section.
Authors: we think that there is a need for a short introduction to the instrumentation used, prior to a more detailed description of the cameras.

Page 4, line 16: What are these "2.52%"? Fraction of the solid angle? Please define.
Authors: We are referring to area only in this sentence: 2.52% is the fraction of the area of the sky. But to remove ambiguity we rephrase the end of the sentence to read: "*2.5% cut off at the top and at the bottom*" (note that now 95%+2.5%+2.5%=100% exactly).

Page 4, line 21: I am not sure how many readers are still familiar with kbaud. One might use kbit/s instead. Authors: we have altered this as suggested.

Page 4, line 25: The shadow disk for the daytime camera is meant to reduce stray light. Isn't the same necessary for the night time camera? The moon is certainly six orders of magnitude darker than the sun, but the lunar halo is as well. Or in other words, the ratio of the brightness lunar halo / moon should be the same as the ratio solar halo / sun. The straylight problem should be the same during day and night.

Authors: The measurements of the halo ratio used the daytime camera, and the night-time camera provided only star positions for testing the lens projection. Hence we show the shade used for the daytime camera only. The referee is probably correct that the stray light effect should be accounted for at night too, but we do not suggest otherwise!

Page 5, line 27: Could you provide the formula of the bi-cubic function and the order of magnitude of the deviation from the ideal camera model? From statements in the following section I infer that it was actually not used? If so, why mention it at all?

Yes, it is correct that the bi-cubic function was not used in the final instance for the correction. To remove the ambiguity, we have replaced the sentences on page 5, lines 25-27:

"*This was found by plotting stars using the preliminary projection and determining the difference between the plotted stars and their corresponding background star. The errors as a function of the x, and then y directions were fitted to a bi-cubic function and these added to the coordinates.*"

with the following:

"*With the preliminary f-theta model so modified the camera parameters were manually adjusted until the difference between the plotted stars and their corresponding background star was minimized.*".

Page 6, line 8: I assume, the test image here was different from the one in the previous section which was used for the geometric calibration?

Authors: Yes, they are not the same. Fig. 2 shows the image used for the testing of the lens projection model, as stated 3 lines later.

Page 7, line 16: Could you briefly (one sentence) explain what vignetting is and how it

is caused?

Authors: the additional text

*Optical and natural vignetting are associated with smaller lens opening for obliquely incident light and the cos4 law of illumination falloff, respectively, both inherent to any lens design (Ray, 2002;"*

has been inserted after

"*...in particular wide-angle and ultra-wide-angle lenses like fisheye lenses*", with the new reference: Ray, S. F.: Applied photographic optics, Focal Press, Oxford, 2002.

Page 7, line 24: Explain acronym LOWESS.

Authors: we have inserted the explanatory text "*(Locally Weighted Scatterplot Smoothing)*".

Page 7, line 27: with "the presence of a peak located roughly $8°$" you mean that the maximum of the function is at $8°$ rather than at $0°$ where you would expect it?

Authors: Yes, that is correct. To clarify the meaning, we have replaced the text with "*the shift of the maximum from the zenith to the position at roughly $8°$*".

Page 8, line 1: Could you please explain why the average between the original and the mirrored curve was used, rather than shifting the curve by $8°$?

Authors: As is explained later in the text, the purpose is to form a rotationally symmetric devingetting function, centred on the zenith.

Figure 8, caption: This is confusing and contradicts the text: The caption says "stretched", while the curve is obviously shifted. The text on the other hand says "mirrored" rather than shifted.

Authors: The red curve is not shifted, it is indeed stretched, i.e. the value at the margin of the image is kept constant. To achieve that, the curve is simply mirrored about the zenith, as explained in the text.

Page 8, line 8: The purpose of the airmass correction is not clear to me. Is it to make the brightness distribution more "flat"?

Authors: The purpose of the air mass correction is simple, as stated at the beginning of section 2.6: the intention is to remove the effect of variable slant path in the single scattering approximation. Otherwise, averaging along lines of constant scattering angle could not be carried out. To clarify this, we have inserted an explanatory sentence "*With this correction in place averaging of sky brightness along lines of constant scattering angle becomes possible.*" after the text "*the corresponding AM(z).*".

Page 8, line 19: To assume single scattering the slant optical thickness has to be smaller than 1. For a viewing zenith angle of 70° the slant optical thickness would be about three times the vertical optical thickness, and thus the assumption of single scattering would only be true for clouds with vertical optical thickness smaller than 0.3. I am sure many of the observed cirrus clouds had a larger optical thickness. Also, the vertically integrated Rayleigh optical thickness in the blue is about 0.2 which adds to the cirrus optical thickness. This once more confirms that you don't observe the scattering phase function but the sky radiance which is caused by cloud scattering plus molecular scattering.

Authors: Please see the first comment for our explanation and justification of our use of the term phase function. Concerning optical thickness, the Reviewer is correct that in some instances a departure from the single scattering approximation is likely have occurred, especially at higher zenith angles. But the purpose of this work was not to obtain the single scattering phase function in the strict sense. The aim was to retrieve the halo ratio, and for that purpose a number of corrections was applied, one of which was the airmass correction, which implicitly assumed the single scattering approximation. So it is a compromise, but one that allowed us to average sky brightness over most of the sky, with the advantages that this brings.

Page 8, line 27: How was the threshold determined empirically? From "manually" deciding if an image is cloud/cloudless and looking at the corresponding fluctuation?

Authors: With reference to Dandini (2016) and Fig. 3.19 and 3.20 therein, the DFA algorithm was applied to one year of continuous BT data. Then the all-sky images

associated with minima of the fluctuation coefficient were selected. From this subset of images a manual cloud screening was done by visual inspection before taking the average FC, which was found to be 0.02. For additional explanation, we now add after "*The DFA algorithm was applied to one year of data and the corresponding FC time series was averaged over clear sky periods to set the DFA threshold.*" the following sentence: "*This was obtained after manual cloud screening of images associated with minima of the fluctuation coefficient.*".

Page 9, line 8: With "irradiance due to the direct emission of cirrus corrected for the atmospheric attenuation" you actually mean after correction to brightness temperature assuming Stephan-Bolthmann? The large deviation from -38°C results from the fact that your broadband instrument integrates over water vapor absorption as well as the atmospheric window? At first glance I expected it to be closer to -38°C, but that would only be the case for a narrowband instrument measuring in the atmospheric window.
Authors: The Stephan-Boltzmann does not apply to this case, as we approximate the narrow-band spectral response of the pyrometer KT15.85 (in the atmospheric window) to be monochromatic, at the central wavelength.
As to the text "*...the irradiance due to the direct emission of cirrus corrected for the atmospheric attenuation, was obtained (see Fig. 10 and 12, middle plot, black dashed line).*", it means that the direct cirrus emission is attenuated by the water vapour layer below the cloud before entering the instrument; this quantity is then added to the modelled water vapour direct emission and the final result is turned into brightness temperature by inverting Planck's law. The large deviation of the Ci threshold from -38° is mainly due to the fact that the model accounts also for direct emission from the water vapour layer below the cloud. For greater clarity we have rephrased the text to read "*...the irradiance due to the direct emission of cirrus corrected for the atmospheric attenuation and emission was obtained, and converted to brightness temperature by inverting Planck's law (see Fig. 10 and 12, middle plot, black dashed line).*".

---

## Author Comment (AC2) · 2 Jul 2018

**Response to Anonymous Referee #1**

We thank the Referee for many insightful comments and helpful suggestions. We list them below, together with our clarifications and changes to the manuscript, where relevant.

First of all, we must point out that Referee #1, perhaps misunderstanding the purpose of the initial, pre-discussion "access review", provided a similar, detailed review prior to the discussion paper being published. In response the manuscript underwent substantial changes and was then published as the discussion paper. However, in several instances in the latter parts of the present comment the Referee has not taken these changes into account and instead appears to refer to the initial manuscript, which unfortunately is not made available as part of the public discussion. This makes our response and its interpretation difficult and potentially confusing, so to help the readers where this occurs we quote the initial review (labelled "Access review, Reviewer") and response (showing the initial changes and labelled "Access review, Authors", in red) followed by the final reply (labelled "RC1, Authors", in red).

RC1, Reviewer:
• Abstract, Page 2, line 32 and especially Page 6, line 20:
The term "scattering phase function" should not be used in this context. It seems that the analysis in this study is based on the measured brightness distribution of the camera which includes multiple scattering rather than the actual scattering phase function, which is a single scattering property.
Access review, Reviewer:
◦ Page 2, line 31 and especially Page 6, line 19:
The term "scattering phase function" is confusing in this context. It seems that the analysis in this study is based on the measured brightness distribution of the camera which includes multiple scattering rather than the actual scattering phase function, which is a single scattering property.
Access review, Authors:
This is intentional: the corrections applied to the measured radiance are intended to provide an approximation to the unnormalized (1,1) element of the scattering matrix. We use the term phase function as shorthand for this property.
RC1, Authors:
We have responded in the access review. We define our meaning of the term phase function early in the text. However, on page 2, after "*Nevertheless, sky imaging is effective for recording the optical displays sometimes associated with cirrus called halos and for measuring the angular distribution of scattered light.*" for greater clarity we continue as follows "*We will henceforth refer to this quantity Scattering Phase Function as the corrections applied to the measured radiance are intended to provide an approximation to the angular dependence of the unnormalized (1,1) element of the scattering matrix*". Further explanation is given in the reply to RC2.

RC1, Reviewer:
· Page 3, line 9:
"...the reflectivity is inversely proportional to the HR." and line 14: "The asymmetry parameter […] is expected to be positively correlated with HR"

These statements are based on observations (Ulanowski et al., 2006; Gayet et al., 2011; Ulanowski et al., 2014) with most HR values < 1, i.e. no 22° halo, which has to be emphasized in this context. Since it is questionable whether differentiating values HR<1 is meaningful at all, the relationship between HR and asymmetry parameter should be explained here first without assumptions:

In general, the reflectivity is primarily determined by the asymmetry parameter which depends on the ice crystal surface roughness and aspect ratio in a U-shaped manner. The HR of the scattering phase function increases for ARs ranging from plates over compact crystals to columns. This implies an ambiguity in the relationship between HR and asymmetry parameter and thus also on the reflectivity.

See

◦ van Diedenhoven 2012: Remote sensing of ice crystal asymmetry parameter using multidirectional polarization measurements – Part 1: Methodology and evaluation with simulated measurements

◦ van Diedenhoven 2014: The prevalence of the 22° halo in cirrus clouds

◦ van Diedenhoven 2014a: A Flexible Parameterization for Shortwave Optical Properties of Ice Crystals

Access review, Authors:

Such ambiguity is indeed expected to exist for geometrically regular, hexagonal ice crystals. However, recent work (see the cited references) indicates that cirrus clouds are dominated by irregular crystals. Ulanowski et al. (2014) show that the halo ratio is inversely proportional to the broadly defined crystal roughness, which in turn is inversely proportional to the asymmetry parameter. As already cited, Gayet et al. (2011) show from in situ observations that the halo ratio is strongly positively correlated with the asymmetry parameter. We now repeat this citation on page 3.

RC1, Authors:

This comment is nearly a verbatim copy of the one given in the access review, too long to be quoted here and to which we believe to have provided adequate response in the initial reply.

· Many abbreviations and acronyms, especially in Sections 2.7 and 3, make the paper difficult to read.

In the following more detailed comments:

1. **Introduction:**

RC1, Reviewer:

The Introduction mainly focuses on cloud detection using all-sky imagers. Are these methods relevant for the presented study?

Access review, Reviewer:

◦ The Introduction mainly focuses on cloud detection using all-sky imagers. Are these methods relevant for the presented study? I would expect a more detailed review of the literature related to camera observations of halo displays, which is only briefly mentioned (Lynch et al., 1985, Sassen etal. 1994, Forster et al. 2017).

Access review, Authors:

No further literature on all-sky cameras observations of halo displays was found. If we take Forster et al. 2017 as the most recent review, aside from some more amateur-like observing networks of volunteers keeping track of halo phenomena as in (Verschure, P.-P., 1998. ) and in (Pekkola, M., 1991), they do not mention any further work in this respect.

RC1, Authors:

We have provided a response in the initial reply to the nearly identical access review. However we can add that it is important to mention the various fields of investigation in which sky imaging finds application: cloud detection is one of these. Also this allows us to

point out widespread use of all-sky cameras and ease with which they could be extended to recording halos and measuring the angular distribution of scattered light.

RC1, Reviewer:
2. **Section 2.2.1:**
◦ Please provide some references for the calibration method using the coordinates of stars.
Access review, Reviewer:
3. **Section 2.2.1:** The description of the calibration method seems a bit incomplete:
◦ Please provide some references for the calibration method, e.g. Kannala and Brandt 2006: "A Generic Camera Model and Calibration Method for Conventional, Wide-Angle, and Fish-Eye Lenses
Access review, Authors:
This is the calibration method for estimating the parameters of the equidistant projection model. This work is not about accurate geometric modelling of real cameras aiming for fine correction of radial and tangential distortions. Described in this section is the calibration procedure to find the camera parameters by assuming the equidistant projection model ($r=f\theta$, "f-theta system"). This assumption is tested in section 2.2.2., and found to be satisfactory.
RC1, Authors:
We have responded to a nearly identical comment in the access review – see above. For clarification, our camera characterization method using the coordinates of stars is not taken from previous work. It is a novel aspect of this study.

RC1, Reviewer:
◦ How does the fit to a bi-cubic function work? Please provide the complete camera model used for the calibration.
Access review, Reviewer:
◦ How does the fit to a bi-cubic function work? Please provide the complete camera model used for the calibration
Access review, Authors:
The only camera model used for the testing of the lens projection is the one identified by eq. 1 and 2, the equidistant projection model.
RC1, Authors:
The bi-cubic function was not used in the final instance for the correction. We have clarified this by replacing the sentences on page 5, lines 25-27: "*This was found by plotting stars using the preliminary projection and determining the difference between the plotted stars and their corresponding background star. The errors as a function of the x, and then y directions were fitted to a bi-cubic function and these added to the coordinates.*" with the following: "*With the preliminary f-theta model so modified the camera parameters were manually adjusted until the difference between the plotted stars and their corresponding background star was minimized.*"

RC1, Reviewer:
◦ Which kind of distortion is corrected – tangential or radial or both?
Access review, Reviewer:
Which kind of distortion is corrected – tangential or radial or both?
Preliminary review, Authors:
There is no distortion correction implemented here. Section 2.2.1 is meant to explain the procedure adopted to measure the camera parameters, namely f, $\Delta$, $x_0$, and $y_0$.
RC1, Authors:
For greater clarity we have replaced the phrase in the Abstract "*geometric correction of lens*

*distortion*" with: "*geometric camera characterization*".

Also, on page 5 we replace "*Finally the empirical scaling factor was applied to the coordinates in the x and y direction to compensate for the distortion created by the lens.*" with "*Finally an empirical scaling factor was applied to the coordinates in the x and y direction. With the preliminary f-theta model so modified the camera parameters were manually adjusted until the difference between the plotted stars and their corresponding background star was minimized.*"

RC1, Reviewer:
◦ How many pictures are used for the calibration?
Access review, Reviewer:
◦ How many pictures are used for the calibration?
Preliminary review, Authors:
While one image was used for the camera calibration, for the testing of the lens projection more than 8000 images were processed, roughly one for each point (blue/red dot, fig. 2) of the star/planet's trajectories.
RC1, Authors:
For greater clarity we add on page 5 after: "*Geometric calibration of the camera is done by detecting the position of specific stars and planets in a night-time image and implementing a minimization procedure.*" the following sentence: "*This was achieved by using four images, taken at different times of night so that bright stars were available in all quadrants of the image.*"

RC1, Reviewer:
◦ What is the effect of increasing the lens aperture on the accuracy of the geometric calibration?
Access review, Reviewer:
What is the effect of increasing the lens aperture on the accuracy of the geometric calibration?
Access review, Authors:
The tests had to be done with the aperture fully open, so there is no way to determine this effect, if present. However, we do not expect it to be significant. Moreover, the errors that were determined must be the worst case, as agreement with the f-theta projection can be expected to be similar or better for the closed aperture.
RC1, Authors:
A reply was provided in the access review.

RC1, Reviewer:
3. **Section 2.2.2:** The authors state a reprojection error of "mostly <0.1° aside from portions of […] trajectories for which larger discrepancies, up to 0.38°, were observed". How large is the error in the region of interest, i.e. where the 22° halo occurs? Is the accuracy sufficient for this study?
Access review, Reviewer:
4. **Section 2.2.2:** The authors state a reprojection error of "mostly <0.1° aside from portions of […] trajectories for which larger discrepancies, up to 0.38°, were observed". How large is the error in the region of interest, i.e. where the 22° halo occurs? Is the accuracy sufficient for this study?
Access review, Authors:
Such error cannot be stated, as the sun, and hence the halo, can be present at a broad range of zenith angles.

RC1, Authors:
A reply was provided in the access review.

4. **Section 2.5:** Using sun photometer measurements to estimate the vignetting effect of the camera is an interesting approach, but some issues are not discussed.
RC1, Reviewer:
◦ Page 7, line 20: What is the spectral response of the blue channel? The influence of Rayleigh scattering in the blue channel is much stronger compared to the red channel and small deviations in the considered wavelengths might cause larger errors in the radiance distribution. Why not use all 3 (RGB) channels? What is the required accuracy of the vignetting correction?
Access review, Reviewer:
◦ Page 8, lines 3-4: What is the spectral response of the blue channel? The influence of Rayleigh scattering in the blue channel is much stronger compared to the red channel and small deviations in the considered wavelengths might cause larger errors in the radiance distribution. Why not use all 3 (RGB) channels? What is the required accuracy of the vignetting correction?
Access review, Authors:
We use the blue channel as it provides the best spectral match with the sun-photometer operating wavelengths and also because it has a larger quantum efficiency when compared to the green and red channels. Furthermore the quantum response of the red sensor has a larger spectral width.
RC1, Authors:
On page 7 for greater clarity we rephrase "*The blue channel of a clear sky daytime RGB image was extracted, being the best match to a spectral channel of the sun photometer.*" as follows: "*The blue channel of a clear sky daytime RGB image was extracted, being the best match to a spectral channel of the sun photometer and having larger quantum efficiency and narrower spectral width of the response than the green and red channels*."

RC1, Reviewer:
◦ Page 7, line 23: "...by neglecting pixels such that z is greater than approximately 20° over the meridian containing the sun...". Why are these pixels neglected?
Access review, Reviewer:
◦ Page 8, line 6: "...by neglecting pixels such that z is greater than approximately 20° over the meridian containing the sun...". Why are these pixels neglected?
Access review, Authors:
Vignetting is assumed to be symmetric under rotations around the camera zenith [$(x_0,y_0)$ or equivalently $z=0°$]. Hence there is no need to use both halves of the principal plane, the one containing the light source ($z<0°$) and the one that does not contain the light source ($z>0°$), to extract the ratio of the two polynomials, one is enough. We decided to use the half such that $z>0°$ and here is why: with respect to fig. 7, pixels such that $z <-24°$ are not taken into account because in this half of the principal plane pixels suitable for measuring the devignetting coefficient are fewer. In fact the camera "sees" the occulting disk while the sun-photometer "sees" the light source, and therefore over the pixels where the occulting disk is located the devignetting coefficient cannot be measured. Nevertheless, pixels such that $z<0°$ but outside the occulting disk region could have been used to take the ratio of the two fitting polynomials. However, this was not done as it was not needed. In fact from the above symmetry argument the devignetting coefficient could be measured using pixels such that $z>-24°$, as we did.
RC1, Authors:

On page 7 for greater clarity we replace *"by neglecting pixels such that z is greater than approximately 20° over the meridian containing the sun"* with *"As vignetting is assumed to be symmetric under rotations around the camera zenith, pixels such that z is greater than approximately 20° over the meridian containing the sun were neglected and ... ."*

RC1, Reviewer:
◦ Page 7, line 27: "...the presence of a peak located roughly 8° from the zenith was investigated...To form a correction function symmetric about the zenith, the original curve was "mirrored" about the zenith.." Based on which assumption? Is this peak is related to the camera or the sun photometer data? Was this peak observed at a different time as well?
Access review, Reviewer:
◦ Page 8, line 11: "...the presence of a peak located roughly 8° from the zenith was investigated...To form a correction function symmetric about the zenith, the original curve was "mirrored" about the zenith.." Based on which assumption? Is this peak is related to the camera or the sun photometer data? Was this peak observed at a different time as well?
Access review, Authors:
The vignetting correction is intended to be generic, so that it could be applied to many other sites using the same combination of camera and lens. Since the correction is nearly rotationally symmetric, and we concluded that the residual asymmetry is the outcome of a small misalignment of the sensor, which is likely to vary between cameras, the generic, symmetric correction can be applied to sites where deriving a camera-specific correction is not possible due to the absence of a sun photometer.
RC1, Authors:
For greater clarity at the end of section 2.5 we add the following: *"The vignetting correction so obtained is intended to be generic. Since the correction is nearly rotationally symmetric, and we have concluded that the residual asymmetry is the outcome of a small misalignment of the sensor, which is likely to vary between cameras, the generic, symmetric correction can be applied to sites where deriving a camera-specific correction is not possible due to the absence of a sun photometer."*

RC1, Reviewer:
◦ Are the measurements interpolated to the same time? How long did the sun photometer scan take?
Access review, Reviewer:
Are the measurements interpolated to the same time? How long did the sun photometer scan take?
Access review, Authors:
The sun-photometer principal plane measurement is performed every hour, over the various wavelengths, one at a time, about 35 s apart.
RC1, Authors:
On page 7 for greater clarity we replace *"The image brightness along the principal plane (containing the zenith and the sun) and the corresponding sun photometer output were compared;"* with the following sentence: *"The sun-photometer measurement along the solar principal plane, performed every hour, over the various wavelengths, one at a time, about 35 s apart, were compared to the corresponding image brightness from the closest camera measurement,;"*

RC1, Reviewer:
◦ Is the vignetting correction determined in the principal plane applied to the whole camera image? If yes, under which assumption?

Is the vignetting correction determined in the principal plane applied to the whole camera image? If yes, under which assumption?
Access review, Authors:
See the comment above.
RC1, Authors:
See the comment above.

RC1, Reviewer:
◦ Fig. 6: Why is the calibrated brightness distribution (black dashed line) smoothed compared to the original data (red dashed line)?
Access review, Reviewer:
◦ Fig. 6: Why is the calibrated brightness distribution (black dashed line) smoothed compared to the original data (red dashed line)?
Access review, Authors:
This is due to the fact that the red dashed line is obtained by applying only the geometric correction to the raw data without the inclusion of the background mask. Therefore contamination associated with background objects is not screened out.
RC1, Authors:
At the end of section 2.5 after "*Fig. 6 (black-dashed line) shows the SPF corresponding to the raw image from Fig. 5 when geometric, AM, vignetting and mask corrections are included.*" we add the following sentence: "*The latter removes the contamination associated with objects in the field of view, such as trees.*"

RC1, Reviewer:
**5. Section 2.6:** What is the purpose of the air mass correction? Is the applied method of Rapp- Arraras and Domingo-Santos (2011) applicable to cloudy scenes as well? What is the error in this case?
Access review, Reviewer:
5. **Section 2.5:** What is the purpose of the air mass correction? Is the applied method of Rapp-Arraras and Domingo-Santos (2011) applicable to cloudy scenes as well? What is the error in this case?
Access review, Authors:
The purpose of the air mass correction is obvious: it is intended to remove the effect of variable slant path in the single scattering approximation. And the context is thin cirrus, so by definition the scene is cloudy (but the optical depth is low).
RC1, Authors:
The purpose of the air mass correction is simple, as stated at the beginning of section 2.6: the intention is to remove the effect of variable slant path in the single scattering approximation. Otherwise, averaging along lines of constant scattering angle could not be carried out. On page 8 to clarify this we have inserted an explanatory sentence "*With this correction in place averaging of sky brightness along lines of constant scattering angle becomes possible.*" after the text "*the corresponding AM(z).*".

RC1, Reviewer:
**6. Section 2.7:**
◦ The cirrus BT threshold is based on an optically thick cirrus. 22° halos are only visible in thin cirrus. What is the effect of a decreasing optical thickness on the BT threshold?
Access review, Authors:
We have substantially extended section 2.7, given further detail about the cirrus

discrimination algorithm and highlighted where a further description of it can be found in the literature. Additionally, in Conclusions we have stated that the method has not been tested yet, vis: "*In the future these cirrus discrimination methods should be compared to techniques such as microwave radiometry or lidar which would allow us to assess their relative merit*."
RC1, Authors:
We have responded in preliminary review and have already substantially changed and expanded the manuscript (the changes in section 2.7 comprise 15 lines of text). Testing the BT threshold method, which is only a subsidiary topic in this work, is well beyond the scope of the article, as stated clearly in the manuscript.

RC1, Reviewer:
◦ How sensitive is the presented threshold method on variations of cloud cover in the scene?
RC1, Authors:
See the comment above.

RC1, Reviewer:
◦ The DFA threshold was estimated empirically to 0.02. This value is much lower than the values found by Brocard et al. 2011 (~0.1 for clear sky, ~0.5 for stratiform cirrus, ~1 for broken cirrus), why?
RC1, Authors:
While in Brocard et al. (2011) data were resampled into 5 s bins our data have a time resolution of 1 second. Moreover, while Brocard's FC is calculated over a time interval ranging from 60 to 300 seconds, our FC is fitted over interval lengths ranging from 20 to 60 seconds. In this respect on page 8, line 28, we rephrase "*This threshold was found empirically to be 0.02 on the basis of the DFA output calculated every 20 minutes and over a time scale ranging between 60 and 150 seconds.*" as follows: "*This threshold was chosen empirically to be 0.02 on the basis of the DFA output calculated, unlike by Brocard et al.(2011), every 5 minutes, over data sampled with 1 s resolution and for time intervals ranging between 20 and 60 seconds. The intervals were chosen as the  time range over which  the DFA function was relatively stable and the slope of the function (the FC) was as sensitive as possible to the presence of clouds.*"
Furthermore, we wish to point out that like in Brocard et al. (2011) the initial analysis step of cumulative summation (integration) of the time series was not carried out. We note that cumulative summation of a self-affine series shifts the FC by +1 (Heneghan and McDarby, 2000), so the threshold applied here would have a value close to 1 if a standard DFA procedure was followed. We place this explanation in the main text.

RC1, Reviewer:
**7. Section 3.1:**
◦ Page 9, line 28: "...between 8 and 10 am… Over this time window […] the behaviour of the BT and the solar irradiance suggests that the HR increases with the optical thickness τ (see Fig. 10, middle and bottom plots)" This is not clearly visible in Fig. 10. The HR increases from 8:12 until 8:30 am but clearly decreases towards 10 am. How is the optical thickness derived from the displayed data?
Access review, Reviewer:
◦ Page 9, line 15: "...between 8 and 10 am… Over this time window […] the behaviour of the BT and the solar irradiance suggests that the HR increases with the optical thickness τ (see Fig. 10, middle and bottom plots)" This is not clearly visible in Fig. 10. The HR increases

from 8:12 until 8:30 am but decreases towards 10 am. How is the optical thickness derived from the displayed data?

Access review, Authors:

We are discussing optical thickness changes - in this context it is legitimate to consider relative changes in brightness temperature or solar irradiance (the latter opposite in sign), which is done here.

RC1, Authors:

A reply was provided in the access review.

RC1, Reviewer:

◦ Page 9, line 31: Kokhanovsky 2008 showed that the halo contrast is linearly decreasing with increasing optical thickness since molecular and aerosol scattering are neglected. Only **the radiance distribution is increasing up to τ=3 and decreasing for τ>3.**

Access review, Reviewer:

◦ Page 9, line 18: Kokhanovsky 2008 showed that the halo contrast is linearly decreasing with increasing optical thickness since molecular and aerosol scattering are neglected. Only the radiance distribution is increasing up to τ=3.

Access review, Authors:

From Kokhanovsky it follows that there are two regimes with respect to the behaviour of the 22° halo brightness with respect to cloud optical thickness τ. For τ<3 halo contrast increases with τ. Analogously, we observe cirrus whose optical thickness is increasing as testified by the increase of BT and a decrease in solar irradiance, and correspondingly an increase in the halo ratio.

RC1 review, Authors:

The reviewer is confusing our definition of "halo ratio" with Kokhanovsky's definition of "contrast". The reviewer seems to be referring to Fig. 3 of Kokhanovsky's paper showing that "contrast" decreases with τ. However, our definition of the halo ratio differs from Kokhanovsky's definition of "contrast". With respect to the radiance distribution shown in Fig. 2a and 2b of the same paper, page 2, Kokhanovsky states: "*It follows from the analysis of Fig. 2 that basically there are two regimes with respect to the behaviour of the brightness of the halo located around 22° observation angle with respect to the cloud optical thickness. The first effect is the brightening of halo with τ. This occurs till cloud optical thickness of 3 (see Fig. 2a) both in the internal dark halo circle and also in the bright ring. For values of τ>3, the increase in the optical thickness leads to the decrease of halo central ring brightness.*" The behaviour of Kokhanovsky's transmission functions (Fig. 2a and 2b) suggests that, with respect to our definition of the halo ratio, the ratio of the transmitted light at 23° to the transmitted light at 20° increases with τ for τ<3. Analogously, we observe cirrus whose optical thickness is increasing as testified by the increase of BT and a decrease in solar irradiance, and correspondingly an increase in the halo ratio.

RC1, Reviewer:

◦ Page 10, line 2: "Between about 8 and 8:20 am the HR, mostly <1, shows a maximum and a minimum at about 8:06 and 8:12 am, respectively, when a relatively faint 22° halo is visible"

How can the HR be <1 but the image shows a 22° halo? The 22° halo in the image at 8:54 appears brighter than at 8:06 but the HR is smaller. Why? Is this an effect of averaging over the complete 22° halo scattering angle region? If so, the HR might be primarily a measure of cloud fraction in this case.

Access review, Reviewer:

◦ Page 9, line 21: "Between about 8 and 8:20 am the HR, mostly <1, shows a maximum and a

minimum at about 8:06 and 8:12 am, respectively, when a relatively faint 22° halo is visible"
How can the HR be <1 but the image shows a 22° halo? The 22° halo in the image at 8:54 appears brighter than at 8:06 but the HR is smaller. Why? Is this an effect of averaging over the complete 22° halo scattering angle region?

Access review, Authors:

Indeed - the reviewer has provided the most likely answer to the question, but it would be speculative to comment further.

RC1, Authors:

As to the original suggestion of the reviewer to use the HR for measuring cloud fraction, it should be pointed out that there are far better ways of doing so. Anyway, this is not the aim of this work. The ultimate goal is deriving reliable HR statistics. The reviewer's scepticism over whether HR is sensitive to the halo status of cirrus is not justified by the overall behaviour of the HR seen over the test cases discussed.

RC1, Reviewer:
8. **Section 3.2:**
Page 11, line 11: "... we conjecture that in order to observe the HR maxima measured for the two cases discussed here a certain minimum fraction of smooth hexagonal ice columns had to be present. If multiple scattering were to be accounted for, such a fraction could be an underestimation of the actual one." The second sentence is now redundant.

Access review, Reviewer:

Page 10, line 31: "According to previous radiative transfer calculations (Forster et al., 2017), we conjecture that in order to observe the absolute HR maxima measured for the two cases discussed here a fraction of at least 20% of smooth hexagonal ice columns was present. If Rayleigh and aerosol scattering were to be accounted for, such a fraction could be an underestimation of the actual one." Why 20%? The simulations in Forster et al. 2017 account for Rayleigh and aerosol scattering.

Access review, Authors:

We changed the manuscript and rewritten this sentence as: "*According to previous radiative transfer calculations (Forster et al., 2017), we conjecture that in order to observe the absolute HR maxima measured for the two cases discussed here a certain minimum fraction of smooth hexagonal ice columns had to be present. If multiple scattering were to be accounted for, such a fraction could be an underestimation of the actual one.*"

RC1, Authors:

We removed the redundant phrase "*If multiple scattering were to be accounted for, such a fraction could be an underestimation of the actual one.*"

RC1, Reviewer:
**9. Conclusions:**
◦ Page 11, line 29/30: "Overall the HR is shown to be sensitive to the halo status of cirrus as it is well correlated with halo visibility." Can you provide a quantitative statement?

Access review, Reviewer:

◦ Page 11, line 17: "Overall the HR is shown to be sensitive to the halo status of cirrus as it is well correlated with halo visibility." Can you provide a quantitative statement?

Access review, Authors:

Unfortunately, we do not know of a quantitative measure of halo visibility - it is a subjective one.

RC1, Authors:

We repeat our initial reply.

RC1, Reviewer:
◦ Page 12, line 3-9: Discrimination between cloud and clear sky seems to work well using the halo ratio alone, without additional DFA threshold test. Discrimination between cirrus and broken cumulus or other clouds seems necessary. How (well) does it work?
Access review, Reviewer:
◦ Page 11, line 24-30: Discrimination between cloud and clear sky seems to work well using the halo ratio alone. Discrimination between cirrus and broken cumulus or other clouds seems necessary. How does the presented method perform compared to the methods presented in the Introduction for cloud type discrimination and halo detection? (e.g. Shields et al. 2013, Forster et al. 2017)
Access review, Authors:
It is not possible to answer this question - we have pointed out in the conclusions (page 12 lines 3, 4 and 5) that such method "*should be compared to techniques such as microwave radiometry or lidar which would allow us to assess its relative merit.*"
RC1, Authors:
See the access review Authors' response here and "Section 2.7" above - the same manuscript modifications apply. Thus the required changes are already in the current version of the manuscript.

RC1, Reviewer:
◦ How does the presented method perform compared to the methods presented in the Introduction for cloud type discrimination and halo detection? (e.g. Shields et al. 2013, Forster et al. 2017)
Access review, Reviewer:
How does the presented method perform compared to the methods presented in the Introduction for cloud type discrimination and halo detection? (e.g. Shields et al. 2013, Forster et al. 2017)
Access review, Authors:
See the comment above.
RC1, Authors:
See the access review Authors' response here and "Section 2.7" above - the same manuscript modifications apply. The required changes are already in the current version of the manuscript.

RC1, Reviewer:
◦ Page 12, line 26: "We argue that when the 22° halo was visible a percentage of at least 20% of regular ice crystals had to be present if molecular, aerosol and multiple scattering in addition to surface albedo are accounted for, together contributing to a reduction of the halo contrast." What is the basis of this argument? Please explain.
Access review, Reviewer:
◦ Page 12, line 14: "We argue that when the 22° halo was visible a percentage of at least 20% of regular ice crystals had to be present if molecular, aerosol and multiple scattering in addition to surface albedo are accounted for, together contributing to a reduction of the halo contrast." What is the basis of this argument? Please explain.
Access review, Authors:
Radiative transfer calculations (Forster et al. 2017) not accounting for multiple scattering suggest that roughly 20% is the fraction of smooth particles needed for the halo to be visible. Given that the all-sky camera does not screen out the effect of multiple scattering, it is reasonable to think that such fraction could be larger, as multiple scattering reduces halo visibility.

**RC1, Authors:**
Page 12, line 26, we replaced the following sentence "*We argue that when the 22° halo was visible a percentage of at least 20% of regular ice crystals had to be present if molecular, aerosol and multiple scattering in addition to surface albedo are accounted for, together contributing to a reduction of the halo contrast.*" with: "*We argue that when the 22° halo was visible a percentage of at least 20% of regular ice crystals had to be present.*"
This is based on sensitivity studies of light scattering simulations (Forster et. al., 2017) showing (see Fig. 11 of Forster et. al.) that HR>1 is associated to smooth crystal fraction larger than 20%, as already cited (end of section 3).

**RC1, Reviewer:**
◦ Page 12, line 28: "We have also conjectured that when the HR reached its absolute maxima the fraction of such pristine crystals is likely to have been much larger than 20%." See previous comment.
"The remaining fraction could have been composed of irregularly shaped, complex, rough or small ice crystals." In Sections 3.1 and 3.2 fluctuations of the HR are explained only by variations of the cloud optical thickness, now the argument is based on the ice crystal microphysical properties. This is not consistent.

**Access review, Reviewer:**
◦ Page 12, line 16: "We have also conjectured that when the HR reached its absolute maxima the fraction of such pristine crystals is likely to have been much larger than 20%. The remaining fraction could have been composed of irregularly shaped, complex, rough or small ice crystals." In Sections 3.1 and 3.2 fluctuations of the HR are explained only by variations of the cloud optical thickness, now the argument is based on the ice crystal microphysical properties. This is not consistent.

**Access review, Authors:**
We do not understand the Reviewer's objection: the halo ratio depends on both optical thickness and microphysical properties. We make this point several times in the text. Perhaps the Reviewer was confused by the emphasis in the observation sections on optical thickness, as we have its indirect measures (via solar irradiance, BT etc.) hence we can draw conclusions on its impact on HR, while the later discussion also takes in modelling results that account for microphysics.
**RC1, Authors:**
We can only repeat our initial response.

**RC1, Reviewer:**
◦ Page 13, line 1: should be "...e.g. the presence of sun dogs and the 46° halo indicates the presence of aligned plates and **randomly oriented hexagonal crystals**, respectively."

**Access review, Reviewer:**
◦ Page 12, line 23: "...e.g. the presence of sun dogs and the 46° halo indicates the presence of aligned plates or solid columns, respectively." The 46° halo is formed by randomly oriented hexagonal crystals, not aligned solid columns.

**Access review, Authors:**
We did not intend to say that the 46° halo is caused by aligned columns, but solid columns. Perhaps our meaning will be clearer if we replace the conjunction "or" with "and".
It now reads: "*Furthermore, by extending the method to additional halo displays further information on ice crystal geometry could potentially be obtained - e.g. the presence of sun dogs and the 46◦ halo indicates the presence of aligned plates and solid columns, respectively.*"
**RC1, Authors:**

We have already replied to this and changed the phrasing accordingly; moreover we believe our statement to be more precise than the suggested one.

RC1, Reviewer:
◦ Page 13, line 3: "The utilisation of the all-sky cameras to transform the measured light intensity into the scattering phase function and, on a limited extent, the cirrus detection algorithm, are the particularly novel aspects of this work; this has not been done previously to the best of our knowledge." The term "cirrus detection algorithm" appears only in the Abstract and Conclusions. Where is this algorithm explained? The method described in Section 2.7 is not new and was already published by Brocard et al., 2011.

Access review, Reviewer:
◦ Page 12, line 24: "The utilisation of the all-sky cameras to transform the measured light intensity into the scattering phase function and, on a limited extent, the cirrus detection algorithm, are the particularly novel aspects of this work; this has not been done previously to the best of our knowledge." Which cirrus detection algorithm? The used method was already described in Brocard et al., 2011.

Access review, Authors:
Our cirrus detection algorithm does not rely solely on the method of Brocard et al. but includes thresholding of the level, as opposed to the fluctuations, of brightness temperature.

RC1, Authors:
Section 2.7 has already been substantially extended, further detail about the cirrus discrimination algorithm was given and relevant literature cited, as described in "Section 2.7" above. The Reviewer appears not to have re-read the manuscript.

RC1, Reviewer:
◦ Page 13, line 10: " The combined use of these two methods, allows relatively inexpensive halo observations and the retrieval of information pertaining to ice particles size and texture." How can particle size an texture be retrieved?

Access review, Reviewer:
◦ Page 12, line 31: " The combined use of these two methods, allows relatively inexpensive halo observations and the retrieval of information pertaining to ice particles size and texture." How can particle size an texture be retrieved?

Access review, Authors:
We are not saying that particle size and texture can be retrieved, certainly not individually.

RC1, Authors:
We stand by our initial response.

---

## Referee Report (RR1)

Most comments from the access review have not been addressed to a satisfactory level. This is why, following the decision of the editor from 18 Feb 2018, these were repeated in the review of the discussion paper. I can not recommend the publication of this manuscript unless the following major comments have been addressed, most of which have already been raised in the two previous reviews.

The authors address the individual points of the review almost exclusively in their reply. Instead, these should be addressed in detail in the manuscript in order to provide the reader with all necessary information at a glance.

To illustrate the history of reviews and replies, several examples are provided by highlighting the "Access Review, Reviewer", "Access Review, Authors", "RC1, Reviewer", and "RC1, Authors" in color. The most recent comments on the authors' reply and the manuscript ("Reply Reviewer") are shown in black.

Major comments:

1. **Access review, Reviewer:**
   Page 2, line 31 and especially Page 6, line 19:
   The term "scattering phase function" is confusing in this context. It seems that the analysis in this study is based on the measured brightness distribution of the camera which includes multiple scattering rather than the actual scattering phase function, which is a single scattering property.
   **Access review, Authors:**
   This is intentional: the corrections applied to the measured radiance are intended to provide an approximation to the unnormalized (1,1) element of the scattering matrix. We use the term phase function as shorthand for this property.
   **RC1, Reviewer:**
   Abstract, Page 2, line 32 and especially Page 6, line 20:
   The term "scattering phase function" should not be used in this context. It seems that the analysis in this study is based on the measured brightness distribution of the camera which includes multiple scattering rather than the actual scattering phase function, which is a single scattering property.
   **RC1, Authors:**
   We have responded in the access review. We define our meaning of the term phase function early in the text. However, on page 2, after "Nevertheless, sky imaging is effective for recording the optical displays sometimes associated with cirrus called halos and for measuring the angular distribution of scattered light." for greater clarity we continue as follows "We will henceforth refer to this quantity Scattering Phase Function as the corrections applied to the measured radiance are intended to provide an approximation to the angular dependence of the unnormalized (1,1) element of the scattering matrix". Further explanation is given in the reply to RC2.

   **Reply Reviewer:** As mentioned in both previous reviews, the term "scattering phase function" used in the context of this manuscript is misleading. "Scattering phase function" is a standing term describing single scattering properties of particles. It is used here to describe a quantity, which the authors describe themselves correctly, however only in their reply to RC2, as "sky radiance corrected for the relative airmass contribution". This correction does not yield a single scattering property. Hence using the term "scattering phase function" in this context would mean a re-definition of a standing term.
   Later in the manuscript the acronym SPF refers to scattering phase functions calculated from electromagnetic theory ("...modelling studies of SPFs (Macke et al., 1996; Baran and Labonnote, 2007; Um and McFarquhar, 2010; Liu et al., 2013)..."). The authors' notation suggests that these two quantities are the same. It further raises the impression that the

scattering phase function of ice crystals can easily be obtained by simply accounting for geometric, vignetting and air mass correction neglecting multiple scattering and aerosol as stated in the abstract ("We propose to obtain it from the scattering phase function (SPF) derived from all-sky imaging") and conclusions (page 12, line 25: "The utilisation of the all-sky cameras to transform the measured light intensity into the scattering phase function..."). This causes the reader to raise false expectations. The additional sentence on page 2 line 32 "We will henceforth refer to this quantity Scattering Phase Function as the corrections applied to the measured radiance are intended to provide an approximation to the angular dependence of the unnormalized (1,1) element of the scattering matrix" is in my point of view not sufficient to resolve this issue.

Please replace the term "scattering phase function" where it refers to the camera observations by a more correct description, e.g. "sky radiance corrected for the relative airmass contribution".

2. **Introduction:**

Access review, Authors:

Such ambiguity is indeed expected to exist for geometrically regular, hexagonal ice crystals. However, recent work (see the cited references) indicates that cirrus clouds are dominated by irregular crystals. Ulanowski et al. (2014) show that the halo ratio is inversely proportional to the broadly defined crystal roughness, which in turn is inversely proportional to the asymmetry parameter. As already cited, Gayet et al. (2011) show from in situ observations that the halo ratio is strongly positively correlated with the asymmetry parameter. We now repeat this citation on page 3.

RC1, Reviewer:

Page 3, line 9:

"...the reflectivity is inversely proportional to the HR." and line 14: "The asymmetry parameter [...] is expected to be positively correlated with HR"These statements are based on observations (Ulanowski et al., 2006; Gayet et al., 2011; Ulanowski et al., 2014) with most HR values < 1, i.e. no 22° halo, which has to be emphasized in this context. Since it is questionable whether differentiating values HR<1 is meaningful at all, the relationship between HR and asymmetry parameter should be explained here first without assumptions:

In general, the reflectivity is primarily determined by the asymmetry parameter which depends on the ice crystal surface roughness and aspect ratio in a U-shaped manner. The HR of the scattering phase function increases for ARs ranging from plates over compact crystals to columns. This implies an ambiguity in the relationship between HR and asymmetry parameter and thus also on the reflectivity.

See

◦ van Diedenhoven 2012: Remote sensing of ice crystal asymmetry parameter using multidirectional polarization measurements – Part 1: Methodology and evaluation with simulated measurements

◦ van Diedenhoven 2014: The prevalence of the 22° halo in cirrus clouds

◦ van Diedenhoven 2014a: A Flexible Parameterization for Shortwave Optical Properties of Ice Crystals

RC1, Authors:

This comment is nearly a verbatim copy of the one given in the access review, too long to be quoted here and to which we believe to have provided adequate response in the initial reply.

**Reply Reviewer:**

The cited studies observed mostly HR<1, i.e. no visible 22° halo. However, in this study the authors investigate cases with visible 22° halo (HR>1) as well. Therefore it is important to discuss the relationship between HR, asymmetry factor and reflectivity expected from

theory in addition as suggested already in the previous reviews. Otherwise this suggests an over-simplified relationship.

3. **Section 2.2.1: Camera Calibration**

**Access review, Reviewer:**

3. Section 2.2.1: The description of the calibration method seems a bit incomplete: Please provide some references for the calibration method, e.g. Kannala and Brandt 2006: "A Generic Camera Model and Calibration Method for Conventional, Wide-Angle, and Fish-Eye Lenses

**Access review, Authors:**

This is the calibration method for estimating the parameters of the equidistant projection model. This work is not about accurate geometric modelling of real cameras aiming for fine correction of radial and tangential distortions. Described in this section is the calibration procedure to find the camera parameters by assuming the equidistant projection model ($r=f\theta$, "f-theta system"). This assumption is tested in section 2.2.2., and found to be satisfactory.

**RC1, Reviewer:**

2. Section 2.2.1: Please provide some references for the calibration method using the coordinates of stars.

**RC1, Authors:**

We have responded to a nearly identical comment in the access review – see above. For clarification, our camera characterization method using the coordinates of stars is not taken from previous work. It is a novel aspect of this study.

**Reply Reviewer:**

The camera characterization method using coordinates of stars has been published previously (e.g. Shields et al. 2013: "Day/night whole sky imagers for 24-h cloud and sky assessment: history and overview, Appl. Optics, 52, 1605–1616, doi:10.1364/AO.52.001605, 2013" and others). Please add some references to the manuscript.

4. **Section 2.2.1: Camera Calibration**

**Access review, Reviewer:**

Which kind of distortion is corrected – tangential or radial or both?

**Access review, Authors:**

There is no distortion correction implemented here. Section 2.2.1 is meant to explain the procedure adopted to measure the camera parameters, namely f, $\Delta$, x 0 , and y 0 .

**RC1, Reviewer:**

Which kind of distortion is corrected – tangential or radial or both?

**RC1, Authors:**

For greater clarity we have replaced the phrase in the Abstract "geometric correction of lens distortion" with: "geometric camera characterization".
Also, on page 5 we replace "Finally the empirical scaling factor was applied to the coordinates in the x and y direction to compensate for the distortion created by the lens." with "Finally an empirical scaling factor was applied to the coordinates in the x and y direction. With the preliminary f-theta model so modified the camera parameters were manually adjusted until the difference between the plotted stars and their corresponding background star was minimized."

**Reply Reviewer:** Instead of removing information in lines 25-28 on page 5, please state in the manuscript that lens distortion was not considered for the geometric calibration.

5. **Section 2.5: Vignetting correction**

**Access review, Reviewer:**

Page 8, lines 3-4: What is the spectral response of the blue channel? The influence of Rayleigh scattering in the blue channel is much stronger compared to the red channel and small deviations in the considered wavelengths might cause larger errors in the radiance distribution. Why not use all 3 (RGB) channels? What is the required accuracy of the vignetting correction?

**Access review, Authors:**

We use the blue channel as it provides the best spectral match with the sun-photometer operating wavelengths and also because it has a larger quantum efficiency when compared to the green and red channels. Furthermore the quantum response of the red sensor has a larger spectral width.

**RC1, Reviewer:**

Page 7, line 20: What is the spectral response of the blue channel? The influence of Rayleigh scattering in the blue channel is much stronger compared to the red channel and small deviations in the considered wavelengths might cause larger errors in the radiance distribution. Why not use all 3 (RGB) channels? What is the required accuracy of the vignetting correction?

**RC1, Authors:**

On page 7 for greater clarity we rephrase "The blue channel of a clear sky daytime RGB image was extracted, being the best match to a spectral channel of the sun photometer." as follows: "The blue channel of a clear sky daytime RGB image was extracted, being the best match to a spectral channel of the sun photometer and having larger quantum efficiency and narrower spectral width of the response than the green and red channels."

**Reply Reviewer:**

What is the basis of stating that the blue channel of the RGB image is the "best match to a spectral channel of the sun photometer" and has a "narrower spectral width of the response than the green and red channels" (page 7, line 24)? Please provide supporting information in the manuscript.

What is the error introduced by Rayleigh scattering? The RGB camera might detect a larger contribution of Rayleigh scattering than the sun-photometer data due to the different spectral width of their channels. Please provide information in the manuscript.

6. **2.5 Vignetting correction:**

The procedure how the vignetting correction was determined and some justifications are still not clearly explained in the manuscript. The following list is a summary of several points, which were raised already in the previous reviews:

- "The sun-photometer measurement along the solar principal plane, performed every hour..." Please state how many sun-photometer measurements were used to determine the vignetting correction.
- Please provide the corresponding camera image which was used for calculating the vignetting correction.
- Did all data (also from different days) exhibit the shift of the peak by about 8° as shown in Fig. 8?
- Was a geometric average of both sun-photometer channels (0.5015 µm and 0.4403 µm) used for the correction? This might also introduce an error since they should be weighted according to the spectral response of the blue channel of the RGB camera.
- Page 8, line 11/12: AM is not yet defined here. The authors might consider moving this sentence to end of Section 2.6 (Air mass correction).
- "...when geometric, AM, vignetting and mask corrections are included. The latter removes the contamination associated with objects in the field of view, such as trees"

does not explain why the curve in Fig. 6 (black dashed line) is smoothed for sca>40°. If a mask was applied, the measurements close to the horizon should be excluded, not smoothed. In Section 2.4 (Background mask) the authors state that "the measure was limited to images such that the light source z<65°". Does this mean that only images for a solar zenith angle <65° were considered? Should the background mask not rather constrain the scattering angles after the image has been transformed and rotated into the zenith as in Fig. 5?

- Page 8, line 14: "Since the correction is nearly rotationally symmetric, and we have concluded that the residual asymmetry is the outcome of a small misalignment of the sensor, which is likely to vary between cameras..." If the shift in the peak of the vignetting correction is due to a misalignment of the camera this offset of 8° should be accounted for in the geometric calibration. Then, the vignetting correction should be determined using the geometrically calibrated image.

- It is not clear how the black dashed line in Fig. 8 was determined from the "original" and the "stretched" (should be "mirrored" like in the text?) curve? It might also seem intuitive to take the "envelope" of the "original" (red) and "mirrored" black curve as vignetting correction.

- With all these approximations it is questionable whether the vignetting correction provided here can serve as a generic correction to other sites as stated on page 8, lines 14-17.

7. **2.6 Air mass correction:**
   The following question from the Access Review and RC1 has not been answered:
   - Is the applied method of Rapp-Arraras and Domingo-Santos (2011) applicable to cloudy scenes as well? What is the error in this case?

   In addition, the following points remain unclear:
   - Which equation from Rapp-Arraras and Domingo-Santos 2011 was used in this study? Where does the value "40" (page 8, line 22) come from?
   - It is not clear what is meant by "the kink" described in line 24 and how it is removed. Please add a figure to illustrate.

8. **Section 2.7: Cirrus discrimination**
   **Access review, Authors:**
   We have substantially extended section 2.7, given further detail about the cirrus discrimination algorithm and highlighted where a further description of it can be found in the literature. Additionally, in Conclusions we have stated that the method has not been tested yet, vis: "In the future these cirrus discrimination methods should be compared to techniques such as microwave radiometry or lidar which would allow us to assess their relative merit."
   **RC1, Reviewer:**
   6. Section 2.7:
   The cirrus BT threshold is based on an optically thick cirrus. 22° halos are only visible in thin cirrus. What is the effect of a decreasing optical thickness on the BT threshold?
   **RC1, Authors:**
   We have responded in preliminary review and have already substantially changed and expanded the manuscript (the changes in section 2.7 comprise 15 lines of text). Testing the BT threshold method, which is only a subsidiary topic in this work, is well beyond the scope of the article, as stated clearly in the manuscript.

   **RC1, Reviewer:**
   How sensitive is the presented threshold method on variations of cloud cover in the

**RC1, Authors:**
See the comment above.

**Reply Reviewer:** The authors have not addressed the questions from RC1:
→ What is the effect of a decreasing optical thickness on the BT threshold?
→ How sensitive is the presented threshold method on variations of cloud cover in the scene?

9. **3.1 Test case 1: 6ᵗʰ July 2016**

   **Access review, Reviewer:**
   Page 9, line 15: "...between 8 and 10 am... Over this time window [...] the behaviour of the BT and the solar irradiance suggests that the HR increases with the optical thickness τ (see Fig. 10, middle and bottom plots)" This is not clearly visible in Fig. 10. The HR increasesfrom 8:12 until 8:30 am but decreases towards 10 am. How is the optical thickness derived from the displayed data?

   **Access review, Authors:**
   We are discussing optical thickness changes - in this context it is legitimate to consider relative changes in brightness temperature or solar irradiance (the latter opposite in sign), which is done here.

   **RC1, Reviewer:**
   7. Section 3.1: Page 9, line 28: "...between 8 and 10 am... Over this time window [...] the behaviour of the BT and the solar irradiance suggests that the HR increases with the optical thickness τ (see Fig. 10, middle and bottom plots)" This is not clearly visible in Fig. 10. The HR increases from 8:12 until 8:30 am but clearly decreases towards 10 am. How is the optical thickness derived from the displayed data?

   **RC1, Authors:**
   A reply was provided in the access review.

   **Reply Reviewer:** The relative changes described in the text (page 10, lines 15-18) do not match the results shown in Fig. 10. How was the optical thickness determined? The points from the Access Review and RC1 were not addressed in the authors' reply.

10. **3 Results and discussion**
    Page 10, lines 6-10: When comparing the scattering angles used to calculate the HR the wavelength or spectral channel used in the respective study should be considered. Which channel is used in this study to compute the HR?

11. **3.1 Test case 1: 6ᵗʰ July 2016**

    **Access review, Reviewer:**
    Page 9, line 18: Kokhanovsky 2008 showed that the halo contrast is linearly decreasing with increasing optical thickness since molecular and aerosol scattering are neglected. Only the radiance distribution is increasing up to τ=3.

    **Access review, Authors:**
    From Kokhanovsky it follows that there are two regimes with respect to the behaviour of the 22° halo brightness with respect to cloud optical thickness τ. For τ<3 halo contrast increases with τ. Analogously, we observe cirrus whose optical thickness is increasing as testified by the increase of BT and a decrease in solar irradiance, and correspondingly an increase in the halo ratio.

    **RC1, Reviewer:**
    Page 9, line 31: Kokhanovsky 2008 showed that the halo contrast is linearly decreasing with

increasing optical thickness since molecular and aerosol scattering are neglected. Only the radiance distribution is increasing up to τ=3 and decreasing for τ>3.

**RC1 review, Authors:**

The reviewer is confusing our definition of "halo ratio" with Kokhanovsky's definition of "contrast". The reviewer seems to be referring to Fig. 3 of Kokhanovsky's paper showing that "contrast" decreases with τ. However, our definition of the halo ratio differs from Kokhanovsky's definition of "contrast". With respect to the radiance distribution shown in Fig. 2a and 2b of the same paper, page 2, Kokhanovsky states: "It follows from the analysis of Fig. 2 that basically there are two regimes with respect to the behaviour of the brightness of the halo located around 22° observation angle with respect to the cloud optical thickness. The first effect is the brightening of halo with τ. This occurs till cloud optical thickness of 3 (see Fig. 2a) both in the internal dark halo circle and also in the bright ring. For values of τ>3, the increase in the optical thickness leads to the decrease of halo central ring brightness." The behaviour of Kokhanovsky's transmission functions (Fig. 2a and 2b) suggests that, with respect to our definition of the halo ratio, the ratio of the transmitted light at 23° to the transmitted light at 20° increases with τ for τ<3. Analogously, we observe cirrus whose optical thickness is increasing as testified by the increase of BT and a decrease in solar irradiance, and correspondingly an increase in the halo ratio.

**Reply Reviewer:** The citation of Kokhanovsky (2008) on page 10, line 18 is not correct: "...the HR increases with the optical thickness τ (see Fig. 10, middle and bottom plots). This is consistent with the results of simulations (Khokanowsky, 2008) [...] that show a linear increase of halo contrast with increasing τ up to τ =3 and a decrease for τ >3 due to multiple scattering.".

As the authors correctly cite Kokhanovsky (2008) (not Khokanowsky, as in the manuscript) in their reply (RC1) the "halo contrast decreases with τ". The authors cite further correctly that the 22° halo first brightens up to an optical thickness of 3 and then decreases for τ>3 – NOT the halo contrast. However, even if the citation of Kokhanovsky in line 18 was corrected, two differently defined quantities are compared: "halo ratio" and "halo contrast". Both the reviewer and the authors seem to agree in this point. So please correct the sentence in the manuscript accordingly or use a more suitable reference.

12. **3.1 Test case 1: 6th July 2016**

**RC1, Reviewer:**

Page 10, line 2: "Between about 8 and 8:20 am the HR, mostly <1, shows a maximum and a minimum at about 8:06 and 8:12 am, respectively, when a relatively faint 22° halo is visible"

How can the HR be <1 but the image shows a 22° halo? The 22° halo in the image at 8:54 appears brighter than at 8:06 but the HR is smaller. Why? Is this an effect of averaging over the complete 22° halo scattering angle region? If so, the HR might be primarily a measure of cloud fraction in this case.

**RC1, Authors:**
As to the original suggestion of the reviewer to use the HR for measuring cloud fraction, it should be pointed out that there are far better ways of doing so. Anyway, this is not the aim of this work. The ultimate goal is deriving reliable HR statistics. The reviewer's scepticism over whether HR is sensitive to the halo status of cirrus is not justified by the overall behaviour of the HR seen over the test cases discussed.

**Reply Reviewer:** Please include some discussion in the manuscript about the possible reasons of the mis-match between HR and the visibility of the 22° halo in the images. The authors seem to have misunderstood RC1. The reviewer did not intend to suggest using HR as a measure of cloud fraction. Instead the reviewer intended to point out that the information content of the HR, which is calculated by averaging over the whole azimuth angle range, might in fact be influenced by changes in the cloud cover. This might be an explanation of the mis-match between HR and the visibility of the 22° halo in the presented case studies.

13. **9. Conclusions**

**Access review, Reviewer:**
Page 11, line 17: "Overall the HR is shown to be sensitive to the halo status of cirrus as it is well correlated with halo visibility." Can you provide a quantitative statement?
**Access review, Authors:**
Unfortunately, we do not know of a quantitative measure of halo visibility - it is a subjective one.
**RC1, Reviewer:**
Page 11, line 29/30: "Overall the HR is shown to be sensitive to the halo status of cirrus as it is well correlated with halo visibility." Can you provide a quantitative statement?
**RC1, Authors:**
We repeat our initial reply.

**Reply Reviewer:** Throughout the manuscript the authors claim the HR as "quantitative measure characterizing the occurrence of the 22° halo peak" (Abstract). So the above mentioned sentence "Overall the HR is shown to be sensitive to the halo status of cirrus as it is **well correlated** with halo visibility" (now page 12, line 17) should be supported by results. This could be done, for example by counting the fraction of images with visible 22° halo and HR>1 vs. the fraction of images with visible 22° halo and HR<1. If it is not possible to support this statement it is not justified.

14. **9. Conclusions**

**Access review, Reviewer:**
Page 12, line 14: "We argue that when the 22° halo was visible a percentage of at least 20% of regular ice crystals had to be present if molecular, aerosol and multiple scattering in addition to surface albedo are accounted for, together contributing to a reduction of the halo contrast." What is the basis of this argument? Please explain.
**Access review, Authors:**
Radiative transfer calculations (Forster et al. 2017) not accounting for multiple scattering suggest that roughly 20% is the fraction of smooth particles needed for the halo to be visible. Given that the all-sky camera does not screen out the effect of multiple scattering, it is reasonable to think that such fraction could be larger, as multiple scattering reduces halo visibility.
**RC1, Reviewer:**
Page 12, line 26: "We argue that when the 22° halo was visible a percentage of at least 20% of regular ice crystals had to be present if molecular, aerosol and multiple scattering in

addition to surface albedo are accounted for, together contributing to a reduction of the halo contrast." What is the basis of this argument? Please explain.

**RC1, Authors:**

Page 12, line 26, we replaced the following sentence "We argue that when the 22° halo was visible a percentage of at least 20% of regular ice crystals had to be present if molecular, aerosol and multiple scattering in addition to surface albedo are accounted for, together contributing to a reduction of the halo contrast." with: "We argue that when the 22° halo was visible a percentage of at least 20% of regular ice crystals had to be present."

This is based on sensitivity studies of light scattering simulations (Forster et. al., 2017) showing (see Fig. 11 of Forster et. al.) that HR>1 is associated to smooth crystal fraction larger than 20%, as already cited (end of section 3).

**Reply Reviewer:** Page 12, line 14-16: "We argue that when the 22° halo was visible a percentage of at least 20% of regular ice crystals had to be present. We have also conjectured that when the HR reached its absolute maxima the fraction of such pristine crystals is likely to have been much larger than 20%. The remaining fraction could have been composed of irregularly shaped, complex, rough or small ice crystals."

Please remove or rephrase these sentences. The results presented in this study do not support that "a percentage of at least 20% regular crystals had to be present". This statement also does not seem to appear anywhere in Forster et al. 2017. They only state 10% as minimum fraction of smooth ice crystals in the conclusions by referring to Diedenhoven 2014.

15. **9. Conclusions**

**Access review, Reviewer:**

Page 12, line 23: "...e.g. the presence of sun dogs and the 46° halo indicates the presence of aligned plates or solid columns, respectively." The 46° halo is formed by randomly oriented hexagonal crystals, not aligned solid columns.

**Access review, Authors:**

We did not intend to say that the 46° halo is caused by aligned columns, but solid columns. Perhaps our meaning will be clearer if we replace the conjunction "or" with "and".

It now reads: "Furthermore, by extending the method to additional halo displays further information on ice crystal geometry could potentially be obtained - e.g. the presence of sun dogs and the 46∘ halo indicates the presence of aligned plates and solid columns, respectively."

**RC1, Reviewer:**

Page 13, line 1: should be "...e.g. the presence of sun dogs and the 46° halo indicates the presence of aligned plates and randomly oriented hexagonal crystals, respectively."

**RC1, Authors:**

We have already replied to this and changed the phrasing accordingly; moreover we believe our statement to be more precise than the suggested one.

Page 13, line 23: As stated in the previous reviews, the sentence as it is phrased now is not correct. Please change it to "...e.g. the presence of sun dogs and the 46° halo indicates the presence of aligned plates or randomly oriented hexagonal crystals, respectively."

**Reply Reviewer:** Repeating the comments from both previous reviews, the statement (now page 13 line 23) is not correct. The 46° halo is caused by randomly oriented hexagonal crystals (plates or columns). The sentence should be corrected as follows: "...the presence of sun dogs and the 46° halo indicates the presence of aligned plates and **randomly oriented hexagonal crystals**, respectively."

Minor comments:

1. At some places in the manuscript it is not clear whether the z refers to the zenith angle in world or camera coordinates.
   In Eq. (1) and (2) on page 6 z represents the azimuth angle in camera coordinates.
   In Eq. (4) on page 7 $\theta$ is defined as the light source zenith angle.
   $\rightarrow$ For example: page 7, line 15: "...the measure was limited to images such that the light source $z < 65°$." If this angle refers to the light source zenith angle, should it be $\theta$ instead of z? Similarly on page 8, line 27 ("Furthermore for large solar z...").

2. Fig. 4: Please explain the the red arrow denoted by "u" in the figure caption.
3. Fig. 5: Which quantity is denoted by "R"? Please explain in the figure caption.
4. Fig. 6: What is "PF"?
5. Fig. 10 and 12: For better readability please explain the acronyms in the figure caption.

---

## Editor Decision (ED1)

Editor: Sorry, no! This paper is not correctly referenced in my opinion.

Hoyningen-Huene call the observed quantity "normalized sky brightness" (as was suggested before by one or both reviewers), see figure 3 of Hoyningen-Huene, and a number of occurrences in their text. The "normalized sky brightness" is defined in their equation (8). Hoyningen-Huene do not assume that this is the scattering phase function. Instead it is stated more than once in the text that the scattering phase function is retrieved using an iterative technique: "For the retrieval of the phase functions from almucantar and spectral AOT the CIRATRA approach (Wendisch and von Hoyningen-Huene, 1994; von Hoyningen-Huene and Posse, 1997) is used." The derived scattering phase function is shown in Figure 4. It looks similar to the normalized sky brightness in the logarithmic diagrams, but not identical. As far as I can see, there is only one occasion in the text where they use the single scattering approximation, and this is in the forward peak of the aureole (which might be justified because the first order scattering contribution is much larger than the higher orders in this narrow angular region).

This is also related to the next point.

Authors:
As we are looking at cirrus the assumption of low optical thickness (i.e. single scattering approximation) is legitimate. This is already stated at page 8, line 18: "In this context we will be assuming single scattering approximation." For greater clarity we reinforce this point in our reply to the reviewer comment two points further down.

Editor: The single scattering approximation is valid for optical thicknesses much smaller than 1 (with an emphasis on „much"), see e.g. Stamnes and Thomas, "Radiative Transfer in the Atmosphere and Ocean", page 232: "This expression shows that it is permissible to use first-order scattering provided that omega0 * tau << 1". To illustrate what that means, I asked an acknoledged world expert in radiative transfer modeling (that is, our model http://www.libradtran.org) which allows separating different orders of scattering using the Monte Carlo approach. I assumed moderate conditions – optical thickness 0.5, solar zenith angle 40°, pristine hexagonal columns, Rayleigh scattering, no aerosol. The results are shown in the figures below. The top figure shows the radiance in the principle plane. Forward scattering peak (at 40°) and the halo are clearly visible. Obviously, in this case only 50% of the signal are caused by single scattering. First and second order scattering account for about 90% of the signal and if one adds the third scattering order then the signal is more or less explained. Since the brightness is normalized anyway, the interesting question is if the ratio of single to multiple scattering shows any structure, which is shown in the 2nd plot. The ratio clearly shows signatures of the halo which implies that the shape of the single

scattering contribution (= phase function) differs from the measured brightness distribution, even if corrected for the airmass. By the way, as indicated above, in the aureole region the signal is much more dominated by single scattering which justifies the above mentioned simplification by Hoyningen-Huene.

Optical thickness of 0.5 and larger is perfectly realistic for a cirrus cloud. Sassen and Comstock, "A Midlatitude Cirrus Cloud Climatology from the Facility for Atmospheric Remote Sensing. Part III: Radiative Properties", JAS 2001 show in their figure 7 that typical midlatitude cirrus clouds have optical thickness between 0 and 2. And when the sun is low in the sky, the slant optical thickness is even larger by a factor 1/cos(SZA)) which is the relevant quantity for the single scattering approximation.

It was therefore suggested before not to use the term "scattering phase function" because it is misleading. The reader most likely will think that the authors actually found a way to derive the scattering phase function of the ice particles (similar to what Wendisch and von Hoyningen-Huene, 1994 did). I (and the reviewers) find that the sentence „We propose to obtain it from the scattering phase function (SPF) derived from all-sky imaging" implies too much – why not simply state what you did, rather than implying assumptions and approximations in the 2nd sentence of the abstract?

It is reasonable, though, to state in the text though, that **if the optical thickness is much smaller than 1** then the airmass-corrected brightness distribution is an approximation of the phase function. But since you do not determine the optical thickness, you do not know if the single scattering approximation does apply or not.

---

## Author Response (AR2)

Halo ratio from ground based all-sky imaging.
*P. Dandini, Z. Ulanowski, D. Campbell, and R. Kaye*
amt-2018-3, submitted on 04 Jan 2018

**Point by point response to revised review**

We thank the Referee for many insightful comments and helpful suggestions. We list them below, together with our clarifications and changes to the manuscript, where relevant.

**Reviewer #1**

**Minor comments**:

**Reviewer**:
Please replace the term "scattering phase function" where it refers to the camera observations by a more correct description, e.g. "sky radiance corrected for the relative airmass contribution".
**Authors:**
We previously approached two acknowledged world experts in different fields to seek their unbiased opinion and they have supported our usage in the present context. We can also cite the following publication, Hoyningen-Huene, W., Dinter, T., Kokhanovsky, A., Burrows, J., Wendisch, M., Bierwirth, E., Müller, D., Diouri, M.: Measurements of desert dust optical characteristic at Porte au Sahara during SAMUM, Tellus B, 61, 206-215, doi:10.1111/j.1600-0889.2008.00405.x., 2009, to further back our usage. We now cite this at page 2, line 30, after: "*We will henceforth refer to this quantity Scattering Phase Function as the corrections applied to the measured radiance are intended to provide an approximation to the angular dependence of the unnormalized (1,1) element of the scattering matrix. This use of the term is consistent with previous practice (e.g. Hoyningen-Huene et al., 2009)."*.

**Reviewer**:
The cited studies observed mostly HR<1, i.e. no visible 22° halo. However, in this study the authors investigate cases with visible 22° halo (HR>1) as well. Therefore it is important to discuss the relationship between HR, asymmetry factor and reflectivity expected from theory in addition as suggested already in the previous reviews. Otherwise this suggests an over-simplified relationship.
**Authors:**
In paragraph 3.2 of Gayet et al. (2011) the authors discuss cirrus properties with 22° halo occurrences, i.e. visible halo, and show that g and HR are positively correlated. The state of theoretical modelling of the halo ratio is at present inadequate. E.g. the work of Diedenhoven previously cited by the Reviewer is based on geometric optics, which does not correctly describe the halo ratio or the asymmetry parameter. Instead, we have referred the readers to the discussion in the works already cited.

**Reviewer**:

The camera characterization method using coordinates of stars has been published previously (e.g. Shields et al. 2013: "Day/night whole sky imagers for 24-h cloud and sky assessment: history and overview, Appl. Optics, 52, 1605–1616, doi:10.1364/AO.52.001605, 2013" and others). Please add some references to the manuscript.

**Authors:**

We already cite Shields et al. 2013. However, Shields et al. do not give any further information regarding calibration beyond stating "Star locations are determined with the aid of a very accurate angular calibration and the use of a bright star catalog". Therefore the citation would be inappropriate in this context.

**Reviewer**:

Instead of removing information in lines 25-28 on page 5, please state in the manuscript that lens distortion was not considered for the geometric calibration.

**Authors:**

For greater clarity we now add at the end of the camera calibration paragraph, the following sentence: *"The camera characterization just described does not account for lens distortion.".*

**Reviewer**:

What is the basis of stating that the blue channel of the RGB image is the "best match to a spectral channel of the sun photometer" and has a "narrower spectral width of the response than the green and red channels" (page 7, line 24)? Please provide supporting information in the manuscript.

**Authors:**

For greater clarity we now add at page 7, line 23, a reference to the camera sensor: *"The blue channel of a clear sky daytime RGB image, with peak wavelength of 0.46 μm, was extracted* (TRUESENSE imaging, KAI-0340 Image Sensor)*, as it provides the best match to the spectral channels of the sun photometer (1.0205, 1.6385, 0.8682, 0.6764, 0.5015, 0.4403 μm) and has larger quantum efficiency and narrower spectral width than the green and red channels."* We also add the reference to the user manual of the sun-photometer at the beginning of the instrumentation paragraph as follows: *"In addition to the all-sky cameras, the Cimel sun photometer CE318 N (Cimel Electonique, 2015) has been necessary to quantify image vignetting.".*

**Reviewer:**

What is the error introduced by Rayleigh scattering? The RGB camera might detect a larger contribution of Rayleigh scattering than the sun-photometer data due to the different spectral width of their channels. Please provide information in the manuscript.

**Authors:**

As already stated in the text, based on the information provided the error introduced by using the red or green channels would be certainly larger than the one associated with the blue channel due to the larger mismatch with the corresponding operating wavelengths of the sun-photometer. A mismatch in the spectral widths of the corresponding channels has little bearing on the error in this context.

**Reviewer:**

6. 2.5 Vignetting correction:

The procedure how the vignetting correction was determined and some justifications are still not clearly explained in the manuscript. The following list is a summary of several points, which were raised already in the previous reviews:

• "The sun-photometer measurement along the solar principal plane, performed every hour..." Please state how many sun-photometer measurements were used to determine the vignetting correction.

• Please provide the corresponding camera image which was used for calculating the vignetting correction.

**Authors:**

For greater clarity we rephrase line 25 page 7: "*The sun-photometer measurement along the solar principal plane, performed every hour, over the various wavelengths, one at a time, about 35 s apart, were compared to the corresponding image brightness from the closest camera measurement;*" as follows: "*A single sun-photometer measurement along the solar principal plane, performed over the various wavelengths, one at a time, about 35 s apart, was compared to the corresponding image brightness from the closest camera measurement (see Fig. 7);*" and add Fig. 7:

[Figure]

**Fig. 7: Bayfordbury all-sky daytime sky image used for vignetting correction, 2/11/2015, 2:57 pm.**

**Reviewer:**

Did all data (also from different days) exhibit the shift of the peak by about 8° as shown in Fig. 8?

**Authors:**

All data exhibit the same shift. We now add at page 8 line 1, after "*As the DC was expected to be monotonically decreasing with z, the shift of the maximum from the zenith to the position at roughly 8° was investigated.*" the following: "*This shift was systematically observed for all data.*" .

**Reviewer:**

Was a geometric average of both sun-photometer channels (0.5015 μm and 0.4403 μm) used for the correction? This might also introduce an error since they should be weighted according to the spectral response of the blue channel of the RGB camera.

**Authors:**

We replace the word "*average*" with the word "*mean*" page 7 line 26.

**Reviewer:**

Page 8, line 11/12: AM is not yet defined here. The authors might consider moving this sentence to end of Section 2.6 (Air mass correction).

**Authors:**

We now define AM on page 8, line 9-10: *"Fig. 6 (black-dashed line) shows the SPF corresponding to the raw image from Fig. 3 when geometric, air mass (AM), vignetting and mask corrections are included.".*

We also change on page 8, line 17: *"To model relative air mass (AM),……"* as *"To model relative AM,……".*

**Reviewer:**

*"...when geometric, AM, vignetting and mask corrections are included. The latter removes the contamination associated with objects in the field of view, such as trees"* does not explain why the curve in Fig. 6 (black dashed line) is smoothed for sca>40°.

**Authors:**

On the contrary, it does explain why the curve in Fig. 6 is smooth; maybe the reviewer is confusing the word "smooth", which we use to refer to the "nice and smooth" behaviour of the curve, with "data smoothing"; there is no data smoothing going on here. If the mask were not there when averaging over the azimuth angle, non-sky pixel brightness, associated with trees, buildings and so forth, would be summed up to sky pixels brightness before taking the mean. We now add for more clarity, page 8, line 10: *"The latter removes the contamination associated with objects in the field of view, such as trees by excluding non-sky pixels from the azimuthal average.".*

**Reviewer:**

If a mask was applied, the measurements close to the horizon should be excluded, not smoothed.

**Authors:**

See above - we do not say anywhere that these measurements are smoothed, the mask is used to exclude "non-sky pixels". There is no smoothing going on here! The mask allows us to remove non-sky pixels from the azimuthal average to which, as a consequence, only the remaining sky pixels contribute.

**Reviewer:**

In Section 2.4 (Background mask) the authors state that "the measure was limited to images such that the light source z<65°". Does this mean that only images for a solar zenith angle <65° were considered?

**Authors:**

For greater clarity we rephrase page 7, line 13: *"the measure was limited to images such that the light source z < 65°."* to: *"only images with $z_{src}$<65° were used.".*

We also rephrase on page 6, line 20:

*"The SPF is obtained by averaging the image brightness over pixels which are equidistant from the light source in terms of scattering angle."*

as follows:

*"The SPF is obtained by averaging the image brightness over pixels which are equidistant from the light source (sun or moon) in terms of scattering angle."*

**Reviewer:**

Should the background mask not rather constrain the scattering angles after the image has been transformed and rotated into the zenith as in Fig. 5?

**Authors:**

The background mask does not constrain the scattering angle at all, that is not its purpose – see above!

**Reviewer:**

Page 8, line 14: "Since the correction is nearly rotationally symmetric, and we have concluded that the residual asymmetry is the outcome of a small misalignment of the sensor, which is likely to vary between cameras..." If the shift in the peak of the vignetting correction is due to a misalignment of the camera this offset of 8° should be accounted for in the geometric calibration. Then, the vignetting correction should be determined using the geometrically calibrated image.

**Authors:**

This work is not about fine calibration of a camera, or lens models. The reviewer fails to understand that the actual aim of this work is providing a method that allows the measurement of the scattering phase function and hence the halo ratio and that permits taking advantage of the large field of view associated with the all-sky imaging in a quantitative manner, overcoming the need of very expensive and unnecessarily sophisticated sun-tracking camera systems. We have explained in the text that the reason for this specific way to correct for vignetting is to provide a "generic" correction, even for sites that are not equipped with sun photometers. We can't be any clearer.

**Reviewer:**

It is not clear how the black dashed line in Fig. 8 was determined from the "original" and the "stretched" (should be "mirrored" like in the text?) curve? It might also seem intuitive to take the "envelope" of the "original" (red) and "mirrored" black curve as vignetting correction.

**Authors:**

The black dashed curve is simply the mean of the original and the mirror curve. For greater clarity we replace the word "*average*" with the word "*mean*" on page 8 line 5. We also replace "*stretched*" with "*mirrored*" in the legend of Fig. 8 (now Fig. 9 after having added the figure Fig. 7).

**Reviewer:**

With all these approximations it is questionable whether the vignetting correction provided here can serve as a generic correction to other sites as stated on page 8, lines 14-17.

**Authors:**

The vignetting correction provided here can indeed serve as a generic correction to other sites. This paper suggests one way of doing this that could, in principle, be improved with finer, individual camera calibration, but at the expense of losing its generic character. However, the novelty of our approach stands despite the unreasonable and unjustified attempts to belittle it. If the reviewer has a vested interest in promoting a different method, this should be declared.

**Reviewer:**

7. 2.6 Air mass correction:

The following question from the Access Review and RC1 has not been answered:

Is the applied method of Rapp-Arraras and Domingo-Santos (2011) applicable to cloudy scenes as well?

**Authors:**

As we are looking at cirrus the assumption of low optical thickness (i.e. single scattering approximation) is legitimate. This is already stated at page 8, line 18: "*In this context we will be assuming single scattering approximation.*" For greater clarity we reinforce this point in our reply to the reviewer comment two points further down.

**Reviewer:**

What is the error in this case?

**Authors:**

We are assuming that such error is negligible. For greater clarity we include this in our reply to the next point.

**Reviewer:**

In addition, the following points remain unclear:

Which equation from Rapp-Arraras and Domingo-Santos 2011 was used in this study? Where does the value "40" (page 8, line 22) come from?

**Authors:**

We rephrase page 8 line 20: *"This provides realistic values of AM near the horizon, where it is less than 40 as it is expected (Rapp-Arraras and Domingo-Santos, 2011)."* as follows: *"This provides realistic values of AM near the horizon as expected (see table 1 in Rapp-Arraras and Domingo-Santos, 2011). Eq 7 from the same work was used. Such functional form has already been used for atmospheres with elevated aerosol layers (Vollmer and Gedzelman, 2006).".*

**Reviewer:**

It is not clear what is meant by "the kink" described in line 24 and how it is removed. Please add a figure to illustrate.

**Authors:**

This is very simple. We now rephrase the text following page 8 line 22 to make it even clearer: *"However, a kink in the SPF, at a scattering angle corresponding approximately to the edge of the mask, was observed."* as follows: *"However, a sudden drop in the value of the SPF towards large scattering angles was observed, ascribable to the large value that the AM takes for large zenith angles, and causing the image brightness to drop rapidly. By excluding such pixels this unwanted drop is significantly reduced. Moreover, it has been shown previously through radiative transfer calculations accounting for multiple scattering that the brightness of the lower part of the halo reaches a maximum for smaller cloud optical thickness $\tau$ than the portion above the sun (Gedzelman and Vollmer, 2008). Consequently, for low solar elevations a zenith cut-off angle that is too large can cause the HR to decrease due to multiple scattering affecting the part of the halo below the sun. By setting the cut-off at z=70° the SPF becomes smooth and multiple scattering effects are reduced, while avoiding seasonal bias due to the solar $z_{src}$ range covered during the year.".*

**Reviewer:**

The authors have not addressed the questions from RC1:

What is the effect of a decreasing optical thickness on the BT threshold?

**Authors:**

We have already responded to this point in two previous responses stating that this is beyond the scope of the present study. Reviewer's comment and authors' response, from RC1, are given below:

"RC1, Reviewer:

The cirrus BT threshold is based on an optically thick cirrus. 22° halos are only visible in thin cirrus. What is the effect of a decreasing optical thickness on the BT threshold? How sensitive is the presented threshold method on variations of cloud cover in the scene?

RC1, Authors:

We have responded in preliminary review and have already substantially changed and expanded the manuscript (the changes in section 2.7 comprise 15 lines of text). Testing the BT threshold method, which is only a subsidiary topic in this work, is well beyond the scope of the article, as stated in the manuscript."

However, for unequivocal clarity we now add, page 9, at the end of paragraph 2.7 what follows:

"*Validation of the presented Ci threshold method is beyond the scope of the present study but will be the subject of a future publication.*" .

**Reviewer:**

How sensitive is the presented threshold method on variations of cloud cover in the scene?

**Authors:**

Such a sensitivity study has not been done – see the previous point.

**Reviewer:**

The relative changes described in the text (page 10, lines 15-18) do not match the results shown in Fig. 10. How was the optical thickness determined? The points from the Access Review and RC1 were not addressed in the authors' reply.

**Authors:**

BT is sensitive to cloud optical thickness and it increases with $\tau$. This makes our statement perfectly legitimate without the need to determine $\tau$.  But we now add for greater clarity, page 10, line 16:

"*Over this time window, on qualitative grounds, the behaviour of the BT and the solar irradiance, which in general grows and decreases with τ, respectively, suggests that the HR increases with the optical thickness τ (see Fig. 10, middle and bottom plots).*" .

**Reviewer:**

10. 3 Results and discussion

Page 10, lines 6-10: When comparing the scattering angles used to calculate the HR the wavelength or spectral channel used in the respective study should be considered. Which channel is used in this study to compute the HR?

**Authors:**

We use the mean of the three channels because the grayscale image is more generic than a single channel, since most all-sky cameras of the type employed here are black and white. We now add at page 10, line 13, the following sentence: "*The SPF was obtained by taking the mean of the three camera channels. This was done because most all-sky cameras of the type described here are greyscale.*".

**Reviewer:**

The citation of Kokhanovsky (2008) on page 10, line 18 is not correct:

"...the HR increases with the optical thickness τ (see Fig. 10, middle and bottom plots). This is consistent with the results of simulations (Khokanowsky, 2008) [...] that show a linear increase of halo contrast with increasing τ up to τ =3 and a decrease for τ >3 due to multiple scattering.".

As the authors correctly cite Kokhanovsky (2008) (not Khokanowsky, as in the manuscript) in their reply (RC1) the "halo contrast decreases with τ". The authors cite further correctly that the 22° halo first brightens up to an optical thickness of 3 and then decreases for τ>3 – NOT the halo contrast. However, even if the citation of Kokhanovsky in line 18 was corrected, two differently defined quantities are compared: "halo ratio" and "halo contrast". Both the reviewer and the authors seem to agree in this point. So please correct the sentence in the manuscript accordingly or use a more suitable reference.

**Authors:**

The wording in our paper contains a fairly obvious conceptual shortcut: since the modelling of Kokhanovsky (2008) does not include molecular and aerosol scattering, as is pointed out, at low optical depth an increase in halo brightness is expected to be associated with an increase in halo contrast, hence our assertion. As for the difference between the HR and contrast, the Reviewer's comment is misplaced: the halo contrast varies monotonically and almost linearly with the halo ratio over the range concerned. However, for clarity we replace, page 10, line 20, the phrase "*halo contrast*" with "*halo brightness*".

**Reviewer**:

Please include some discussion in the manuscript about the possible reasons of the mis-match between HR and the visibility of the 22° halo in the images. The authors seem to have misunderstood RC1. The reviewer did not intend to suggest using HR as a measure of cloud fraction. Instead the reviewer intended to point out that the information content of the HR, which is calculated by averaging over the whole azimuth angle range, might in fact be influenced by changes in the cloud cover. This might be an explanation of the mis-match between HR and the visibility of the 22° halo in the presented case studies.

**Authors:**

We include our answer to this question and corresponding changes to the manuscript in our reply to the next point.

**Reviewer**:

Throughout the manuscript the authors claim the HR as "quantitative measure characterizing the occurrence of the 22° halo peak" (Abstract). So the above mentioned sentence "Overall the HR is shown to be sensitive to the halo status of cirrus as it is **well correlated** with halo visibility" (now page 12, line 17) should be supported by results. This could be done, for example by counting the fraction of images with visible 22° halo and HR>1 vs. the fraction of images with visible 22° halo and HR<1. If it is not possible to support this statement it is not justified.

**Authors:**

We already responded that the halo visibility is a subjective measure. As such it is assessed in a subjective manner, which has been done and can be confirmed by interested readers on the basis of the data contained in the paper, or the broader dataset presented in Dandini (2016). However, for clarification, we add, page 11, line 31, *the following: "Manual inspection of the all-sky images allows us to state that when the HR>1 the halo is visible 95% of the time (true positives) while halo visibility associated with HR<1 represents only 10% of the occurrences (false negatives). This relatively minor fraction may be associated with locally larger values of optical thickness, as the partial cirrus thickening observed in the sky images at 1 pm and 1:24 pm on the 7[th] of July suggests."* We also add in the Conclusions, page 12, line 15, after: *"Overall the HR is shown to be sensitive to the halo status of cirrus as it is well correlated with halo visibility"* the following: *", aside from a relatively minor fraction of the data with visible halo and HR<1, possibly associated with locally larger values of optical thickness.".*

**Reviewer**:

Page 12, line 14-16: "We argue that when the 22° halo was visible a percentage of at least 20% of regular ice crystals had to be present. We have also conjectured that when the HR reached its absolute maxima the fraction of such pristine crystals is likely to have been much larger than 20%. The remaining fraction could have been composed of irregularly shaped, complex, rough or small ice crystals." Please remove or rephrase these sentences. The results presented in this study do not support that "a percentage of at least 20% regular crystals had to be present". This statement also does not seem to appear anywhere in Forster et al. 2017. They only state 10% as minimum fraction of smooth ice crystals in the conclusions by referring to Diedenhoven 2014.

**Authors:**

We rephrase page 13, line 11: *"We argue that when the 22° halo was visible a significant percentage of regular ice crystals had to be present and that such a fraction is likely to have been much larger when the HR reached its maxima. The remaining fraction could have been composed of irregularly shaped, complex, rough or small ice crystals.".*

**Reviewer**:

Repeating the comments from both previous reviews, the statement (now page 13 line 23) is not correct. The 46° halo is caused by randomly oriented hexagonal crystals (plates or columns). The sentence should be corrected as follows: "...the presence of sun dogs and the 46° halo indicates the presence of aligned plates and randomly oriented hexagonal crystals, respectively."

**Authors:**

The Reviewer's statement is wrong: the 46° halo is NOT caused by hexagonal plates, for at least two reasons we can think of! However, for greater clarity we can rephrase *"The presence of sun dogs and the 46° halo indicates the presence of aligned plates and solid columns, respectively."* as follows: *"...the presence of sun dogs and the 46° halo indicates the presence of aligned plates and non-aligned, solid hexagonal prisms, respectively.".*

**Minor comments**:

**Reviewer**:

At some places in the manuscript it is not clear whether the z refers to the zenith angle in world or camera coordinates. In Eq. (1) and (2) on page 6 z represents the azimuth angle in camera coordinates. In Eq. (4) on page 7 θ is defined as the light source zenith angle. For example: page 7, line 15: "...the measure was limited to images such that the light source z < 65°." If this angle refers to the light source zenith angle, should it be θ instead of z? Similarly on page 8, line 27 ("Furthermore for large solar z...").

**Authors:**

Page 6, after eq. 1 and 2 it reads *"where (z , A) are the zenith and azimuth angles, Δ is the rotation of the camera from north, measuring about 16.4° and 13.6° for the night and day time cameras, respectively, $(x_0, y_0)$ are the pixel coordinates of the actual zenith and f is the scale factor."* z is therefore clearly defined as the zenith angle in the sky coordinates; it is never the azimuth angle as the Reviewer states! The use of the phrase "light source" simply allows for generality, as it includes the case of the moon instead of the sun. However, for greater clarity we simplify page 7, line 2, *"by an angle ϑ equal to the light source zenith angle....."* as follows: *"by an angle ϑ equal to the light source zenith angle $z_{src}$ .....".*

We now add $z_{src}$ in Fig. 4, next to the red dashed line linking the light source centre and the zenith, and add the explanation in the corresponding figure caption that now reads: *"Figure 4. Projection of the original image from Fig. 3 onto a sphere (upper hemisphere). The unit vector u determines the direction around which the rotation that leads to the sun-centred image, Fig. 5, takes place. $z_{src}$ is the light source zenith angle."*

**Reviewer**:

Fig. 4: Please explain the the red arrow denoted by "u" in the figure caption.

**Authors:**

We change the figure caption as follows: *"Figure 4. Projection of the original image from Fig. 3 onto a sphere (upper hemisphere). The unit vector u determines the direction around which the rotation that leads to the sun-centred image, Fig. 5, takes place.".*

**Reviewer**:

Fig. 5: Which quantity is denoted by "R"? Please explain in the figure caption.

**Authors:**

We change the figure caption as follows: *"Figure 5. Image from Fig. 4 after rotation (upper hemisphere only). R is the radius of the sphere onto which the original images are mapped.".*

**Reviewer**:
Fig. 6: What is "PF"?
**Authors:**
We replace "PF" with "SPF" in the legends throughout the paper.

**Reviewer:**
Fig. 10 and 12: For better readability please explain the acronyms in the figure caption.
**Authors:**
Fig. 10 and 12 become Fig. 11 and 13, respectively, having added the figure used for the devignetting correction (Fig. 7); we now update the corresponding figure captions by explaining the acronyms as follows:

[revised manuscript text omitted]

---

## Author Response (AR3)

Halo ratio from ground based all-sky imaging.
*P. Dandini, Z. Ulanowski, D. Campbell, and R. Kaye*
amt-2018-3, submitted on 04 Jan 2018

**Response to the Associate Editor's comments**

We thank the Editor for the comments. We list them below, together with our clarifications and changes to the manuscript, where relevant. However, it appears to us that we are now splitting hairs. The review has moved away from the topic of the paper, which is the retrieval of the halo ratio from measured sky brightness, to an abstruse discussion of terminology, namely whether it is legitimate to call the quantity determined in the work an approximation to the "scattering phase function", or whether some other term might be available. This necessitates ever more lengthy explanations of our position, even though it is of minor importance to the context of the work. So we need to state emphatically (again) that the proposed alternative term of "sky brightness" is not appropriate in our case: we do not deal with sky brightness, we deal with an approximation to the scattering phase function. We agree that it is imperfect, but how legitimate is this approximation is a separate issue. Although again it is of minor importance, we discuss this aspect below, using hard data.

**Reviewer:**
Please replace the term "scattering phase function" where it refers to the camera observations by a more correct description, e.g. "sky radiance corrected for the relative airmass contribution".

**Authors:**
We previously approached two acknowledged world experts in different fields to seek their unbiased opinion and they have supported our usage in the present context. We can also cite the following publication, Hoyningen-Huene, W., Dinter, T., Kokhanovsky, A., Burrows, J., Wendisch, M., Bierwirth, E., Müller, D., Diouri, M.: Measurements of desert dust optical characteristic at Porte au Sahara during SAMUM, Tellus B, 61, 206-215, doi:10.1111/j.1600-0889.2008.00405.x., 2009, to further back our usage. We now cite this at page 2, line 30, after: "We will henceforth refer to this quantity Scattering Phase Function as the corrections applied to the measured radiance are intended to provide an approximation to the angular dependence of the unnormalized (1,1) element of the scattering matrix. This use of the term is consistent with previous practice (e.g. Hoyningen-Huene etal., 2009).".

**Editor:**
Sorry, no! This paper is not correctly referenced in my opinion. Hoyningen-Huene call the observed quantity "normalized sky brightness" (as was suggested before by one or both reviewers), see figure 3 of Hoyningen-Huene, and a number of occurrences in their text. The "normalized sky brightness" is defined in their equation (8). Hoyningen-Huene do not assume that this is the scattering phase function. Instead it is stated more than once in the text that the scattering phase function is retrieved using an iterative technique: "For the retrieval of the phase functions from almucantar and spectral AOT the CIRATRA approach (Wendisch and von Hoyningen-Huene, 1994; von Hoyningen-Huene and Posse, 1997) is used." The derived scattering phase function is shown in Figure 4. It looks similar to the normalized sky

brightness in the logarithmic diagrams, but not identical. As far as I can see, there is only one occasion in the text where they use the single scattering approximation, and this is in the forward peak of the aureole (which might be justified because the first order scattering contribution is much larger than the higher orders in this narrow angular region).

This is also related to the next point.

**Authors:**

This comment takes the Hoyningen-Huene et al. (2009) reference out of context, in citing the use of the term "normalized sky brightness". They use almucantar measurements, therefore the airmass is constant, while we deal with variable airmass, and correct for this variability, therefore the term "brightness" is no longer appropriate. The Editor misquotes Hoyningen-Huene et al. (2009) because these authors do not refer to the quantity "normalized sky brightness" in the context of angle-dependent airmass correction (as is the case for us) but merely to sky brightness divided by a constant. This is a crucial distinction, as the latter remains a "brightness" while our quantity is no longer a "brightness" but an approximation to an altogether different quantity, a phase function! Moreover, our usage of the term "phase function" is consistent with the referenced paper in that the CIRATRA approach does not lead to significant differences between the calculated phase function and observed sky radiance in particular up to the region of the 22° halo to which we are mainly interested.

Also, non-cloud-screened AERONET AOT for our site, data corresponding to the test cases discussed in the manuscript shows mostly AOT < 1, see Fig. 1 and 2 below, except at one data point on the 7th of July at about 12:45 when correspondingly our cloud classification algorithm detects warm clouds.

However, for greater clarity page 2 line 32, we replace:

"*This use of the term is consistent with previous practice (e.g. Hoyningen- Huene et al., 2009).*"

with the following:

"*This use of the term is consistent with previous practice (e.g. Hoyningen- Huene et al., 2009; Voltz, 1987) as the air mass corrected sky brightness provides a good approximation to the scattering phase function at least for τ<1. AERONET level 1 optical thickness data corresponding to the test cases discussed confirm that τ<1 except on the 7th of July at about 12:45 when, however, our cloud classification method (see section 2.7) screens out the measurements as associated with warm clouds.*"

[Figure]

*Illustration 1: SDA AOT - 6th of July 2016*

[Figure]

*Illustration 2: Illustration 2: SDA AOT - 7th of July 2016*

**Authors:**

As we are looking at cirrus the assumption of low optical thickness (i.e. single scattering approximation) is legitimate. This is already stated at page 8, line 18: "In this context we will be assuming single scattering approximation." For greater clarity we reinforce this point in our reply to the reviewer comment two points further down.

**Editor:**

The single scattering approximation is valid for optical thicknesses much smaller than 1 (with an emphasis on „much"), see e.g. Stamnes and Thomas, "Radiative Transfer in the Atmosphere and Ocean", page 232: "This expression shows that it is permissible to use first-order scattering provided that omega0 * tau << 1". To illustrate what that means, I asked an acknowledged world expert in radiative transfer modeling (that is, our model http://www.libradtran.org) which allows separating different orders of scattering using the Monte Carlo approach. I assumed moderate conditions – optical thickness 0.5, solar zenith angle 40°, pristine hexagonal columns, Rayleigh scattering, no aerosol. The results are shown in the figures below. The top figure shows the radiance in the principle plane. Forward scattering peak (at 40°) and the halo are clearly visible. Obviously, in this case only 50% of the signal are caused by single scattering. First and second order scattering account for about 90% of the signal and if one adds the third scattering order then the signal is more or less explained. Since the brightness is normalized anyway, the interesting question is if the ratio of single to multiple scattering shows any structure, which is shown in the 2nd plot. The ratio clearly shows signatures of the halo which implies that the shape of the single scattering contribution (= phase function) differs from the measured brightness distribution, even if corrected for the airmass. By the way, as indicated above, in the aureole region the signal is much more dominated by single scattering which justifies the above mentioned simplification by Hoyningen-Huene. Optical thickness of 0.5 and larger is perfectly realistic for a cirrus cloud. Sassen and Comstock, "A Midlatitude Cirrus Cloud Climatology from the Facility for Atmospheric Remote Sensing. Part III: Radiative Properties", JAS 2001 show in their figure 7 that typical midlatitude cirrus clouds have optical thickness between 0 and 2. And when the sun is low in the sky, the slant optical thickness is even larger by a factor 1/cos(SZA)) which is the relevant quantity for the single scattering approximation. It was therefore suggested before not to use the term "scattering phase function" because it is misleading. The reader most likely will think that the authors actually found a way to derive the scattering phase function of the ice particles (similar to what Wendisch and von Hoyningen-Huene, 1994 did). I (and the reviewers) find that the sentence „We propose to obtain it from the scattering phase function (SPF) derived from all-sky imaging" implies too much – why not simply state what you did, rather than implying assumptions and approximations in the 2nd sentence of the abstract? It is reasonable, though, to state in the text though, that **if the optical thickness is much smaller than 1** then the airmass-corrected brightness distribution is an approximation of the phase function. But since you do not determine the optical thickness, you do not know if the single scattering approximation does apply or not.

**Authors:**

To cut this overlong discussion short, we are looking at optically thin cirrus, as testified by the observed values of AOT<1, we can reasonably state that our airmass-corrected brightness distribution is an approximation to the phase function.

However, for greater clarity we add page 8, line 19, after:
"*In this context we will be assuming single scattering approximation.*"
the following:
"*In this context we will be assuming single scattering approximation, as justified by co-located non-cloud-screened AERONET measurements showing optical thickness <1.*"

Moreover, since much of the aforementioned discussion hinges on our inclusion of the airmass correction in our data, we add in the main text, page 8 line 19 after: "*
[revised manuscript text omitted]

---

## Author Response (AR4)

Halo ratio from ground based all-sky imaging.
*P. Dandini, Z. Ulanowski, D. Campbell, and R. Kaye*
amt-2018-3, submitted on 04 Jan 2018

**Response to the Associate Editor's comments**

We thank the Editor for the comments. We transcribe it below, together with our clarifications and changes to the manuscript, where relevant.

Associate Editor:

Dear authors,

we could go on like this forever. However, it seems that is is your explicit wish to use imprecise terminology, so be it!

You are perfectly right in stating that the use of the term "scattering phase function" is of minor importance for your work. I would go even further: It has zero relevance for your work. Hence, why introduce an approximation if it is not needed? Why would one state that PI could be approximated by 4 in the introduction if PI wasn't used at all in a publication?

In your response you clearly state that the topic of your paper is the "retrieval of the halo ratio from measured sky brightness". I couldn't agree more! Two lines down you "state emphatically (again) that the proposed alternative term of "sky brightness" is not appropriate in our case". Why not? Dividing the measured quantity by the cosine of the viewing angle doesn't automagically transform it into the scattering phase function. Since we agree that the approximation is imperfect (your words), why not simply avoid it? My main concern is future papers citing Dandini et al as proof that one can use the single scattering approximation for cirrus clouds of optical thickness around 1. And of course my name as an editor being associated with it.

Thank you for providing (for the first time) numbers for the optical thickness. They are indeed smaller than 1 most of the time, but the slant optical thickness is definitely not MUCH smaller than 1 which is the requirement for the single scattering approximation. One also needs to consider that AERONET severely underestimates the optical thickness of cirrus by a factor between 1 and 2 (see e.g. https://www.atmos-meas-tech.net/12/169/2019/amt-12-169-2019.pdf page 178). Considering this correction (factor 1 .. 2) plus the slant path (factor 1.5) one can conclude that the slant optical thickness is between 1 and 2 most of the time. Even if you quietly replaced the "<<" which is the real requirement for the single scattering approximation by a simple "<", this condition would not be fulfilled most of the time.

In my view, this is evidence enough to NOT use the terms "single scattering approximation" and "scattering phase function" in that context. In particular, since it is not needed at all for the paper.

Go ahead as you please!

Non-public comments to the Author:
Please change it or leave it as it is if you think it is appropriate. A simple "no" is sufficient.

Authors:

We agree that it is time to close this discussion. Like the Editor, we would not wish to be associated with the use of an incorrect term (i.e. sky brightness), that is why we stressed in the text that we were dealing with "*approximation to the scattering phase function*" and further that we were "*assuming single scattering approximation*". To emphasize this further we now add the word "approximation" in the following sentence in the abstract:

[revised manuscript text omitted]